# Structures of complete HIV-1 TAR RNA portray a dynamic platform poised for protein binding and structural remodeling

Charles Bou-Nader [1], Katie A. Link [2], Krishna C. Suddala [1], Jay R. Knutson[2] & Jinwei Zhang [1]✉

The HIV-1 TAR RNA plays key roles in viral genome architecture, transcription and replication. Previous structural analyses focused on its upper stem loop, which has served as a paradigm to study RNA structural dynamics. However, an imperfectly paired lower stem immediately abuts and stacks with the upper half, both of which are required for efficient HIV replication. Here, we report crystal structures of the full-length HIV-1 TAR which reveal substantial conformational mobility in its three conserved bulges and in its lower stem, which coordinately maintain the structural fluidity of the entire RNA. We find that TAR RNA is a robust inhibitor of PKR, and primarily uses its lower stem to capture and sequester PKR monomers, preventing their dimerization and activation. The lower stem exhibits transient conformational excursions detected by a ligation assay. Time-resolved fluorescence spectroscopy reveals local and global TAR structural remodeling by HIV-1 nucleocapsid, Tat, and PKR. This study portrays the structure, dynamics, and interactions of a complete TAR RNA, uncovers a convergent RNA-based viral strategy to evade innate immunity, and provides avenues to develop antivirals that target a dynamic, multifunctional viral RNA.

The HIV-1 Trans-Activation Response element (TAR) RNA, located at both ends of its RNA genome, is one of the most conserved RNA structural elements in retroviruses[1]. It assumes multiple, dynamic conformations to carry out critical functions in several stages of the HIV-1 life cycle, including reverse transcription, proviral transcription, genome dimerization and packaging, viral translation, and immune suppression, etc[2–5].

During early reverse transcription, the minus-strand strong-stop transfer step relocates the initial reverse transcriptase (RT) complex from the 5′ end to the 3′ end of the viral genome[6]. This transfer is mediated by the unwinding and annealing of the 3′ TAR and polyA RNA hairpins with their DNA complement in the RT complex and is facilitated by the chaperone activity of HIV-1 nucleocapsid (NC)[7–9]. In subsequent proviral transcription, HIV-1 trans-activator protein (Tat) binds the UCU-bulge of TAR, recruits positive transcription elongation

factor b (P-TEFb), and assembles the processive super elongation complex (SEC) to transcribe the entire viral genome[10]. Therefore, TAR facilitates both the minus-strand transfer by RT and HIV transcription elongation.

Proviral transcription of the HIV-1 and other retroviral genomes can initiate from any of the three guanosines (Gs) at the 5′ end of TAR and is cotranscriptionally capped, resulting in substantial transcription start site (TSS) heterogeneity[11,12]. This variable number of 5′ Gs on TAR controls the conformation and fate of the 5′ leader and entire genome, in part by modulating TAR interactions with the immediately adjacent polyA hairpin[11,13]. While leaders bearing a single 5′ G are preferably packaged into virions, 3 G leaders are selectively retained in cells and serve as mRNAs for translation[14]. In addition, both types of leaders bear the 7-methylguanosine (m7G) cap, which can further stack with the 5′ G(s) in an inverted geometry. While the capped 3 G leader extrudes and

[1]Laboratory of Molecular Biology, National Institute of Diabetes and Digestive and Kidney Diseases, Bethesda, MD, USA. [2]Laboratory of Advanced Microscopy and Biophotonics, National Heart, Lung, and Blood Institute, Bethesda, MD, USA. ✉e-mail: jinwei.zhang@nih.gov

exposes its 5′ cap for translation initiation factor (eIF4E) binding and translation, the capped 1 G leader does not bind eIF4E under similar conditions, directing it away from ribosomes[13]. The lower, proximal stem of TAR coordinates with the 5′ TSS and the cap to control the overall leader conformation and its fate. In agreement, disruption of the TAR hairpin causes aberrant genome dimerization and packaging[15]. While the hairpin structure of TAR suppresses ribosomal scanning, a TAR-polyA coaxial stacking interaction within the capped 1 G leader additionally conceals the cap from eIF4E, thus reinforcing the translation block[13].

TAR also interacts with a litany of host proteins to modify gene expression and suppress immune responses[16]. TAR directly binds TAR-binding protein (TRBP)[17], PKR[18–20], RNA Helicase A[21], NELF[22], NF90[23], La antigen[24], etc. Of particular interest is PKR, a central component of the interferon-mediated antiviral response[18,25]. The latent kinase activity of PKR is activated by double-stranded RNAs (dsRNAs) produced during viral replication or bidirectional transcription. Activated PKR exerts broad-spectrum antiviral and antiproliferative effects, including the shutoff of viral protein synthesis by phosphorylating the eukaryotic translation initiation 2 (eIF2α) and activation of apoptotic pathways[26,27]. Essentially all known viruses have evolved mechanisms to escape or antagonize PKR. HIV-1 employs a multi-pronged approach to prevent PKR activation by its dsRNA-rich viral genome, involving Tat, TRBP, ADAR1, PACT, and potentially also TAR[28]. Early work suggested that in vitro transcribed TAR activates PKR[19,29] while other studies showed TAR inhibits PKR[30–32]. Recent analyses of T7 RNA polymerase transcription products revealed its surprising propensity to utilize product RNA as templates to generate dsRNA byproducts[33,34]. Trace amounts of such dsRNA are sufficient to robustly activate PKR[33]. Thus, the precise effect of TAR on PKR remains unclear.

Pioneering structural studies of HIV-1 TAR produced a wealth of general insights into the conformational dynamics of RNA bulges and hairpins[35–40]. Nonetheless, these have primarily focused on the upper stem-loop of TAR (nucleotides (nts) 18–44; "mini-TAR", or "ΔTAR"), largely due to technical limitations such as significant line broadening of resonances in nuclear magnetic resonance (NMR) analysis of longer RNAs[4]. However, the 27-nt mini-TAR resides within a ~ 57-nt-long, continuous hairpin structure. The lower stem immediately abuts and stacks coaxially with the upper stem-loop, and is expected to exert strong influences on the conformational equilibria and kinetics of the upper stem-loop. The lower stem can also stack with the adjacent polyA hairpin, creating a strong interdependency between the conformations of both hairpins[13]. In accordance, the lower stem of TAR contributes to Tat-mediated transcriptional activation[41,42] and is required for optimal HIV-1 replication[43]. Laboratory evolution analysis revealed that lower stem mismatches completely blocked HIV-1 replication and revertant viruses rescued replication by restoring the pairing in the lower stem[43]. These observations highlight the critical importance of the conserved lower stem of TAR, hint at a conformational coupling between the two halves of the TAR hairpin, and prompted our structural analyses of the full-length TAR RNA. Herein we report four crystal structures of this 57-nt RNA, both free and bound to HIV-1 Tat. Ligation probing analyses corroborate the crystal structures in solution and reveal a metastable lower stem of TAR that is chiefly responsible for PKR binding and inhibition. Besides PKR, we also characterized TAR interactions with HIV-1 Tat and NC and employed time-resolved fluorescence spectroscopy to observe how various proteins differentially remodel the TAR structure.

## Results

### The proximal half of HIV-1 TAR anchors PKR binding and inhibition

To ascertain HIV-1 TAR's effect on PKR, we first performed PKR activation and inhibition assays. Activation assays showed that highly purified full-length TAR RNA does not activate PKR across a wide range of RNA concentrations (1–1000 nM), on par with the Adenovirus Virus-Associated RNA I (VA-I), a well-characterized, highly potent PKR inhibitor (Fig. 1a, b). Further, when pre-bound with PKR, TAR blocked kinase activation by outcompeting a 79-base pair (bp) dsRNA − a strong PKR activator (Fig. 1c, e and Supplementary Table 1). HIV-1 TAR inhibited PKR with an apparent $IC_{50}$ of ~ 77 nM, compared to a ~ 11 nM $IC_{50}$ by VA-I. PKR dimerization upon dsRNA binding is considered necessary for its activation[18,27,30,44]. Size exclusion chromatography coupled to multi-angle light scattering (SEC-MALS) revealed 1:1 stoichiometric binding of PKR to TAR (Supplementary Fig. 1a, b), similar to VA-I[44]. The lack of PKR dimerization on TAR affirms that TAR, like VA-I, is a robust inhibitor of the kinase, and sequesters it in a monomeric state.

We next mapped which region of TAR is responsible for PKR binding and inhibition. Previous EDTA•Fe cleavage and imino proton NMR analyses suggested that both dsRNA-binding motifs (dsRBMs) of PKR engage both the upper and lower stems of TAR[19,20]. To understand the relative contribution of these two regions, we progressively truncated the terminal region of TAR by 4, 9, or 14 bp and measured binding by full-length PKR or its dsRBMs using fluorescence polarization (FP, Fig. 1a, d, e, Supplementary Fig. 1c, d, g–j and Supplementary Table 2). We found that a 4-bp truncation had a minor impact on binding while the 9-bp and 14-bp truncations reduced PKR-binding affinity by ~ 4 and ~ 35-folds, respectively. These findings suggest that the lower stem is necessary for high-affinity binding. Remarkably, the lower stem alone, capped by a tetraloop ($TAR^{\Delta23-39}$), exhibited near-WT binding affinities. These results suggest that the lower stem of TAR is the primary PKR-binding site (Fig. 1d, e and Supplementary Fig. 1g, j).

We then examined PKR inhibition by these truncation variants. Contrasting the gradual, progressive loss of PKR binding, PKR inhibition is abolished in all three terminal truncation constructs, including the short 4-bp deletion (Fig. 1c, e, asterisks). Remarkably, the capped lower stem ($TAR^{\Delta23-39}$) retained most of its PKR-inhibition ability, exhibiting only a 5-fold larger $IC_{50}$. Therefore, the lower stem is the most important TAR element for both PKR binding and kinase inhibition. Consistent with this, opening up the lower stem by three mismatches in the lower stem ($TAR^{A48U/C49G/U50A}$) reduced PKR binding by 3-fold and kinase inhibition by 22-fold (Fig. 1c, e and Supplementary Fig. 1k).

Helical imperfections, mismatches, and bulges in long dsRNAs are known to reduce PKR activation[32,45]. We next asked if the three conserved bulges (C5, A17, and U23-C24-U25) contribute to PKR binding or inhibition. Deleting the two single-nt bulges ($TAR^{\Delta C5\Delta A17}$, green) had little impact, whereas removing the UCU bulge ($TAR^{\Delta23UCU25}$, brown) reduced PKR inhibition potency by ~ 5-fold (Fig. 1c, e). Interestingly, removing all three bulges ($TAR^{\Delta C5\Delta A17/\Delta23UCU25}$, teal) generated a slightly shorter, near perfectly paired dsRNA, which exhibited 2-fold better binding to PKR and 5-fold enhanced inhibition, achieving comparable potency as VA-I (Fig. 1c, e). These findings suggest that the bulges temper the ability of TAR to recruit and target PKR. The fact that TAR is not optimized for PKR inhibition likely reflects a coordinated evolutionary process during which multiple pro-viral functions collectively shaped the structure, sequence, and stability of TAR. The congruent binding, activation, and inhibition data establish that HIV-1 TAR RNA is a robust PKR inhibitor, and primarily uses its lower stem for PKR binding and inhibition.

Finally, we asked whether the 5′ TSS heterogeneity impacts TAR stability, dynamics, or its PKR inhibition since this heterogeneity modulates the conformation of the entire leader[13]. We compared three TAR constructs: (a) $TAR^{2GC}$ (WT), which bears two Gs on the 5′ end paired to the two Cs on the 3′ end forming blunt termini, (b) $TAR^{3G}$, which has a third, overhanging 5′ G appended to $TAR^{2GC}$, and (c) $TAR^{3GC}$, which has three G-C pairs at the blunt termini (Supplementary Fig. 2a–c). Using Differential Scanning Calorimetry (DSC) and Circular Dichroism (CD), we found that all three full-length TAR RNAs

 

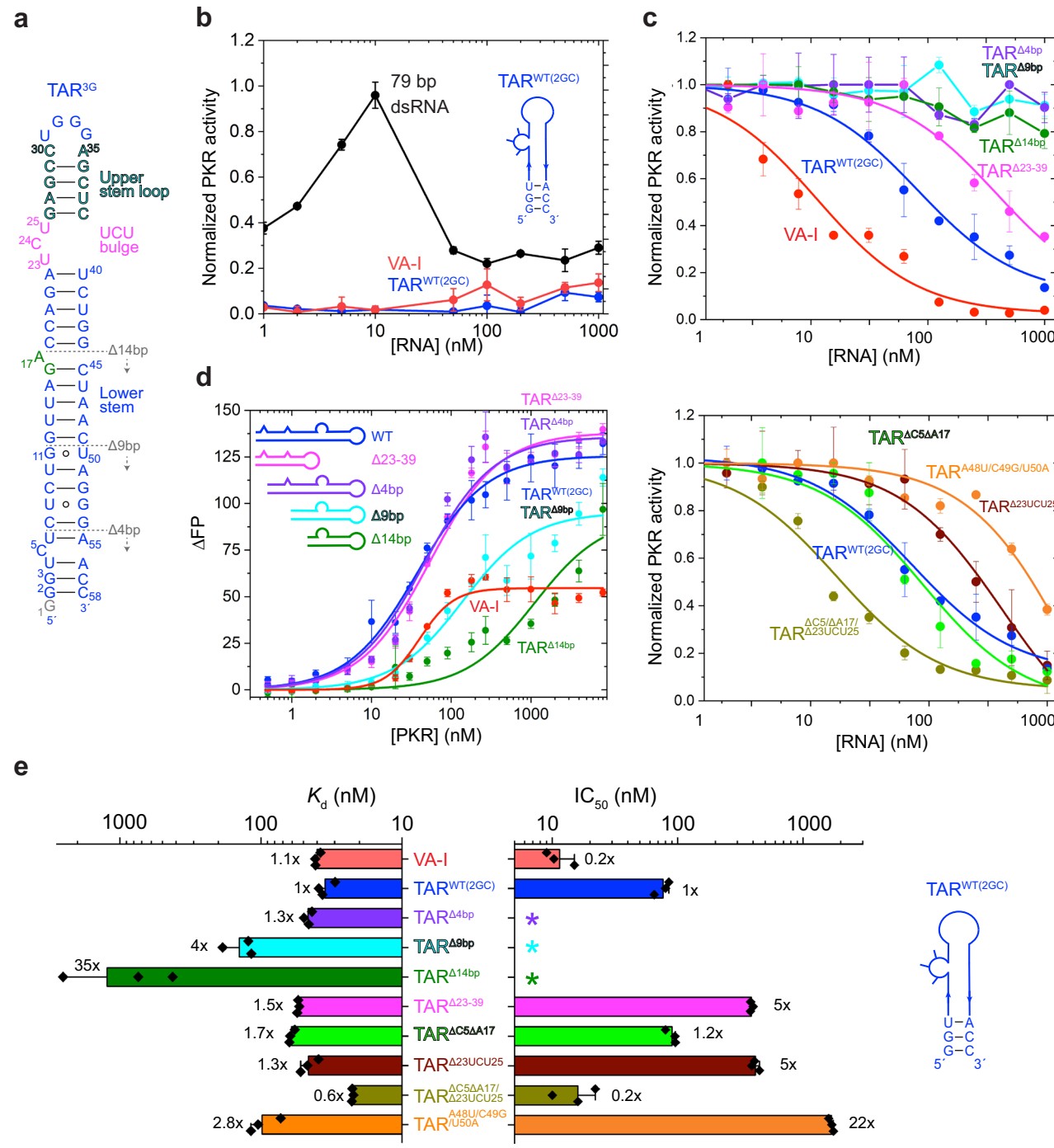

**Fig. 1 | Inhibition of PKR by full-length HIV-1 TAR. a** Secondary structure of complete HIV-1 TAR^3G. G1 is indicated in gray as it is omitted in most TAR constructs used in this study unless explicitly indicated otherwise. The boundaries of TAR truncations are indicated by dotted lines. **b** PKR kinase activation profiles by a 79-bp dsRNA (black), VA-I RNA (red), and HIV-1 TAR^WT(2GC) (blue, inset). **c** PKR inhibition profiles by VA-I (red), TAR^WT(2GC) (blue), TAR^Δ4bp (purple), TAR^Δ9bp (cyan), TAR^Δ14bp (dark green), TAR^Δ23-39 (magenta), TAR^ΔC5ΔA17 (light green), TAR^ΔC5ΔA17Δ23UCU25 (olive), TAR^Δ23UCU25 (brown) and TAR^A48U/C49G/U50A (orange). **d** Fluorescence polarization titration of FAM-labeled VA-I and TAR constructs by increasing concentrations of PKR. **e** Comparison of $K_d$ (left) and IC_{50} (right) values derived from data in (**c** and **d**). Values are mean ± s.d. of three biologically independent samples. *: no significant inhibition observed. Source data are provided in this paper.

predominantly unfolded via single, cooperative transitions. This indicates that the upper and lower halves of the TAR RNA are conformationally coupled, and thermal melting is processive once initiated. The melting temperature ($T_m$) of TAR^2GC was generally increased by ~1 °C by the addition of the third G (in TAR^3G), and by another ~1 °C by completing a third, terminal G-C pair (in TAR^3GC), both in the presence or absence of Mg^{2+} (Supplementary Fig. 2 and Supplementary Table 3). This suggests that the TAR terminal region makes

measurable contributions to its overall thermostability. Interestingly, the addition of a single, unpaired 5′ G guanosine (in TAR^3G) appreciably stabilized the TAR structure, presumably by cross-strand stacking with the adjacent base pair. To ask if the differences in $T_m$ due to TSS heterogeneity can meaningfully impact the dynamic behavior of TAR, we compared the rates of NC-mediated annealing of these three TAR constructs to their complementary TAR DNA (cTAR), during a step in minus-strand transfer where TAR RNA hairpin opens up and hybridizes

to cTAR (Supplementary Fig. 3). Remarkably, we observed a ~ 3-5-fold slowdown of TAR-cTAR annealing in the two slightly longer, more stable TARs (Supplementary Fig. 3a–e). This indicates that the terminal region of WT TAR²ᴳᶜ may be metastable, capable of sampling transiently open conformations that facilitate its base pairing with cTAR. By contrast, all three TARs exhibited comparable inhibitory potency for PKR (Supplementary Fig. 3f), consistent with the notion that dsRBMs bind principally in the internal regions but not the ends of dsRNA.

### Crystal structures of full-length HIV-1 TAR depict a polymorphic, dynamic RNA

To visualize the complete HIV-1 TAR structure and particularly the PKR-binding lower stem, we crystallized the 57-nt RNA by substituting the UGGG portion of the distal hexaloop with a GAAA tetraloop (Supplementary Figs. 4, 5 and Supplementary Table 4). This substitution slightly destabilized the RNA ($T_m$ changed from 78.2 to 77.4 °C, Supplementary Fig. 2m and Supplementary Table 3) and reduced binding of HIV-1 Tat[44–60] (Supplementary Fig. 6r and Supplementary Table 2). This suggests that the distal loop substitution did not drastically perturb the overall TAR structure. These crystals captured two similar conformations in the same unit cell (root mean square deviation (RMSD) of 1.8 Å over 56 C1' atoms) which we term TAR-Ia and TAR-Ib (Fig. 2a, b and Supplementary Fig. 7a, b). We also crystallized a mutant TAR where nts G16 and A17 were swapped at the mid-section (TAR-II, G16A/A17G, GAAA, Fig. 2c), to explore potential effects of shifting the location of the A17 bulge, and viral sequence polymorphism on TAR structure.

All three crystal structures support the predicted general secondary structure of TAR derived from in silico calculations, NMR, mutagenesis, and chemical probing analyses in vitro and in cells[46–48]. The upper stem-loop portion resembles previous NMR and crystal structures of mini-TAR[35,38,39]. As expected, the UCU bulge (nts 23-25) is extrahelical and flexible as evidenced by its poorly defined electron density and elevated temperature factors (Fig. 2d–f, Supplementary Figs. 4a–d and 7c). Interestingly, soaking the TAR-II crystals with CaCl₂ stabilized the extrahelical UCU bulge, and promoted a U23-C24 stacking interaction (Supplementary Fig. 4g). This is apparently mediated by Ca²⁺ coordination with the juxtaposed RNA backbone phosphate oxygens of the bulge. Similar effects of Ca²⁺ were also observed in a previous mini-TAR crystal structure (Supplementary Fig. 4h)[37]. These findings suggest that the distinctive structural, electrostatic, and chemical environment of the UCU bulge, especially its distorted backbone, can create specific cation-binding sites. In turn, binding by divalent cations such as Mg²⁺ and Ca²⁺ can also modulate the conformation of this flexible RNA bulge.

Moving down from the UCU bulge, A17 demarcates the boundary between the upper and lower halves of TAR RNA and was omitted from previous structural studies of mini-TAR. In our full-length TAR structures, A17 is stably flipped out, allowing its two flanking stems to coaxially stack with and stabilize each other. In TAR-II, the G16A/A17G substitutions shifted the bulge one pair down to A16, which similarly permits contiguous stacking through the entire RNA. While the unpaired A17 (or A16) residue located in the mid-section is stably bulged out, the other unpaired residue, C5, located near the termini, assumed three distinct conformations among the three structures (Fig. 2g–i). In TAR-Ia, C5 flips away from the termini and binds in the major groove, forming a base triple with the C9-G52 pair. In TAR-Ib, C5 flips towards the termini and binds the minor groove, forming an inclined base triple with the G3-C57 pair. In TAR-II, C5 is extrahelical and solvent exposed. These observations suggest that single-nt pyrimidine bulges near open dsRNA termini can exhibit considerable mobility and conformational polymorphism. Curiously, the HIV-1 MAL strain employs a TAR element that features one additional single-nt bulge (U8) besides the conserved C5 bulge in the lower stem[14].

Notably, among our structures, none of the three bulges interrupted the coaxial stacking that runs along the entire helical axis, which accentuates the prominent contribution of stacking in shaping RNA structures[49].

Remarkably, the wider, axially compressed helix of TAR overlays nearly perfectly with the coaxially stacked apical stem and tetrastem portions of the Adenovirus VA-I RNA termed mini-VA, which is necessary and sufficient to inhibit PKR (Fig. 2j and Supplementary Fig. 7d, e)[44]. Therefore, different viruses seem to have converged on a common RNA-based strategy to suppress PKR – by using short (24–28 bp), imperfect dsRNA segments to sequester PKR monomers (Fig. 2k). This strategy is likely also shared by certain cellular circular RNAs that present discontinuous dsRNA segments of similar lengths interspersed with bulges to inhibit PKR[50,51].

### HIV-1 Tat[44–60] exhibits an alpha-helical conformation *in crystallo* upon TAR binding

Besides PKR suppression, the full structure of TAR is also essential for Tat-mediated HIV transcriptional activation[41,42]. To understand how a full-length HIV-1 TAR interacts with HIV-1 Tat, we soaked the RNA-binding domain peptide of Tat (Tat[44–60], GISYGRKKRRQRRRAHQ) into TAR-I crystals and solved their complex structure at 2.7 Å resolution (Figs. 3 and 4, Supplementary Fig. 4e, f and Supplementary Table 4). As expected, the Tat[44–60] peptide bound and remodeled the UCU bulge (Fig. 3a, b, e, f). Unexpectedly, most of Tat[44–60] formed an alpha helix upon TAR binding, distinct from a recent NMR structure of a Tat-mini-TAR complex[39] (Fig. 3c–g), but strongly resembles the EIAV (Equine infectious anemia virus) TAR-Tat complex structure[52]. The backbone orientation of the Tat helix is also opposite from the NMR structure but is consistent with earlier proposals based on crosslinking and EPR analyses[53,54]. Notably, reversing the sequence of Tat still permitted TAR binding[55]. The ambiguity in Tat binding directionality on TAR, at least for isolated Tat peptides in vitro, may be attributable to the over-abundance of basic residues (Arg, Lys) in the short peptide (~ 53%), lack of distinguishing sequence features, and its intrinsically disordered protein (IDP)-like behavior.

The inherent tendency for the Tat RNA-binding domain to fold into an alpha helix has been long recognized, which is essential for Tat cell penetration and has inspired the development of cell-penetrating peptides (CPPs)[56]. However, the alpha-helical conformation has not been structurally characterized, and it is also unknown if this conformation is involved in TAR binding. To address this, we first monitored the secondary structure of Tat in solution under the conditions of TAR binding and crystal soaking using CD. As expected, Tat remains unstructured under both conditions (Supplementary Fig. 8a). By titrating in 2,2,2-Trifluoroethanol (TFE), a co-solvent which stabilizes protein alpha helices and other secondary structures, Tat gradually gained an alpha-helical character (~ 15%), as evidenced by the appearance of signature negative bands at 208 and 222 nm (Supplementary Fig. 8b). Due to strong CD signals from RNA, it is not feasible to accurately estimate Tat secondary structure while bound to TAR. Instead, we asked how modulating the helicity of the Tat peptide may impact TAR binding affinities. Alanine and proline substitutions can effectively promote and impede the tendency to form alpha helices in proteins, respectively[57]. We substituted three Tat residues (G48, Q54, A58/Q60) not involved in RNA binding with alanines or prolines. As expected, the triple alanine substitution (G48A/Q54A/Q60A) strongly promoted helix formation while the triple proline substitution (G48P/Q54P/A58P) blocked it, even in the presence of 70% TFE (Supplementary Fig. 8c–e). Both substitutions, despite exerting strong effects on the helicity of the Tat peptide, had little impact on TAR binding (Fig. 4h, Supplementary Fig. 6 and Supplementary Table 2). These findings suggest that the alpha-helical conformation of Tat, required for cell surface penetration, is not necessary for TAR recognition in solution. Tat can adopt an alpha-helical fold when bound to TAR, at

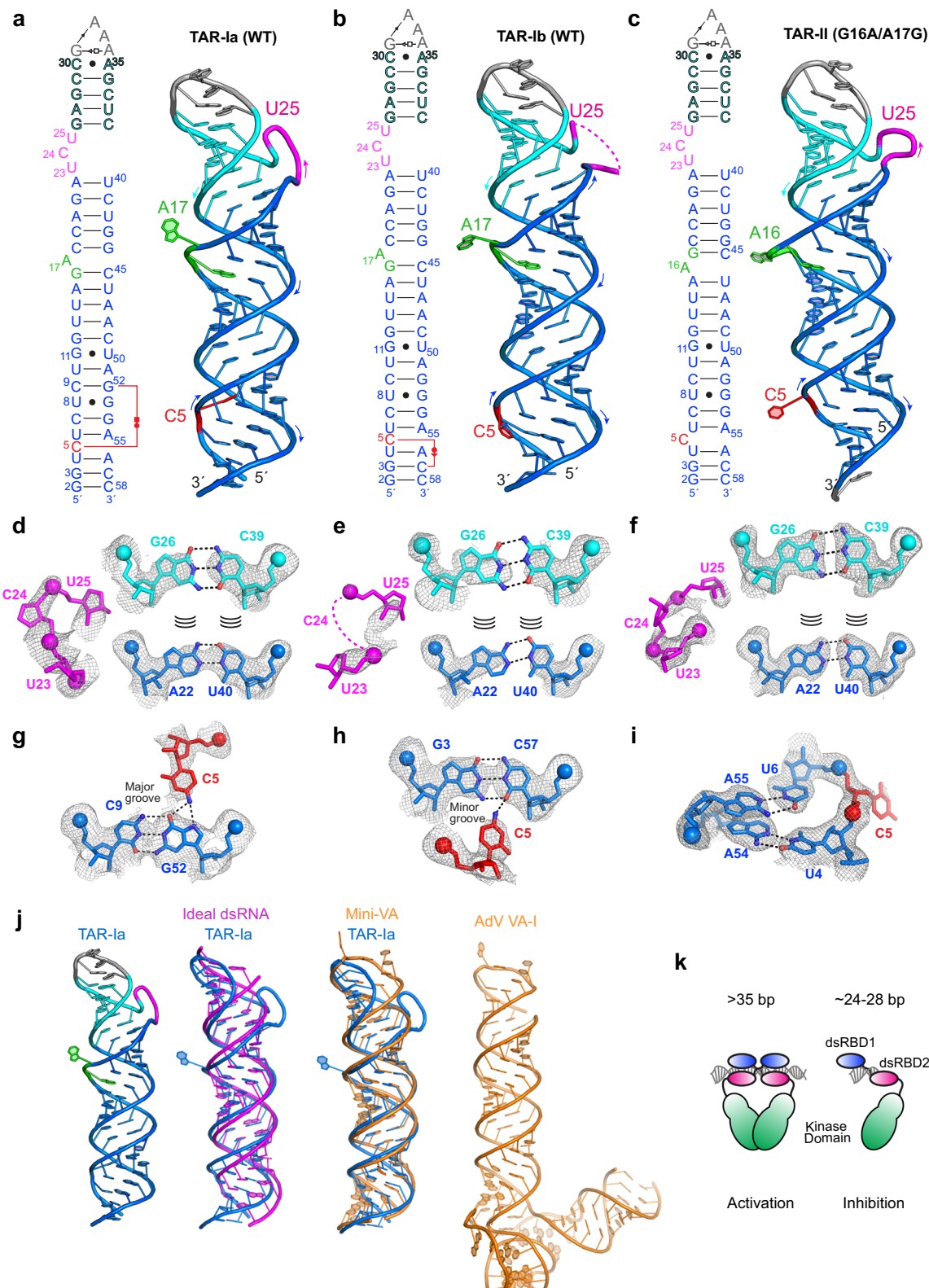

**Fig. 2 | Crystal structures of full-length HIV−1 TAR. a−c** Overall and secondary structures of TAR-Ia (**a**), TAR-Ib (**b**), and TAR-II (**c**). Non-canonical pairs are indicated by Leontis-Westhof symbols[91]. **d−f** Zoomed-in views of the UCU bulge and two flanking pairs overlayed with the 2Fo·Fc electron density contoured at 1σ for TAR-Ia (**d**), TAR-Ib (**e**) and TAR-II (**f**). Dashed lines represent proposed hydrogen bonds; curve lines indicate nucleobase stacking; spheres denote phosphorus atoms. **g−i** Zoomed-in views of the C5 region with the 2Fo·Fc electron density contoured at 1σ for TAR-Ia (**g**), TAR-Ib (**h**), and TAR-II (**i**). **j** Structural overlay of TAR-Ia (blue) with an ideal A-form dsRNA (magenta) and the mini-VA (fused apical stem-tetrastem of VA-I RNA, orange, PDB: 6OL3[44]). A full-length VA-I structure (orange) is shown for comparison. **k** Model of PKR regulation by dsRNA of varying lengths.

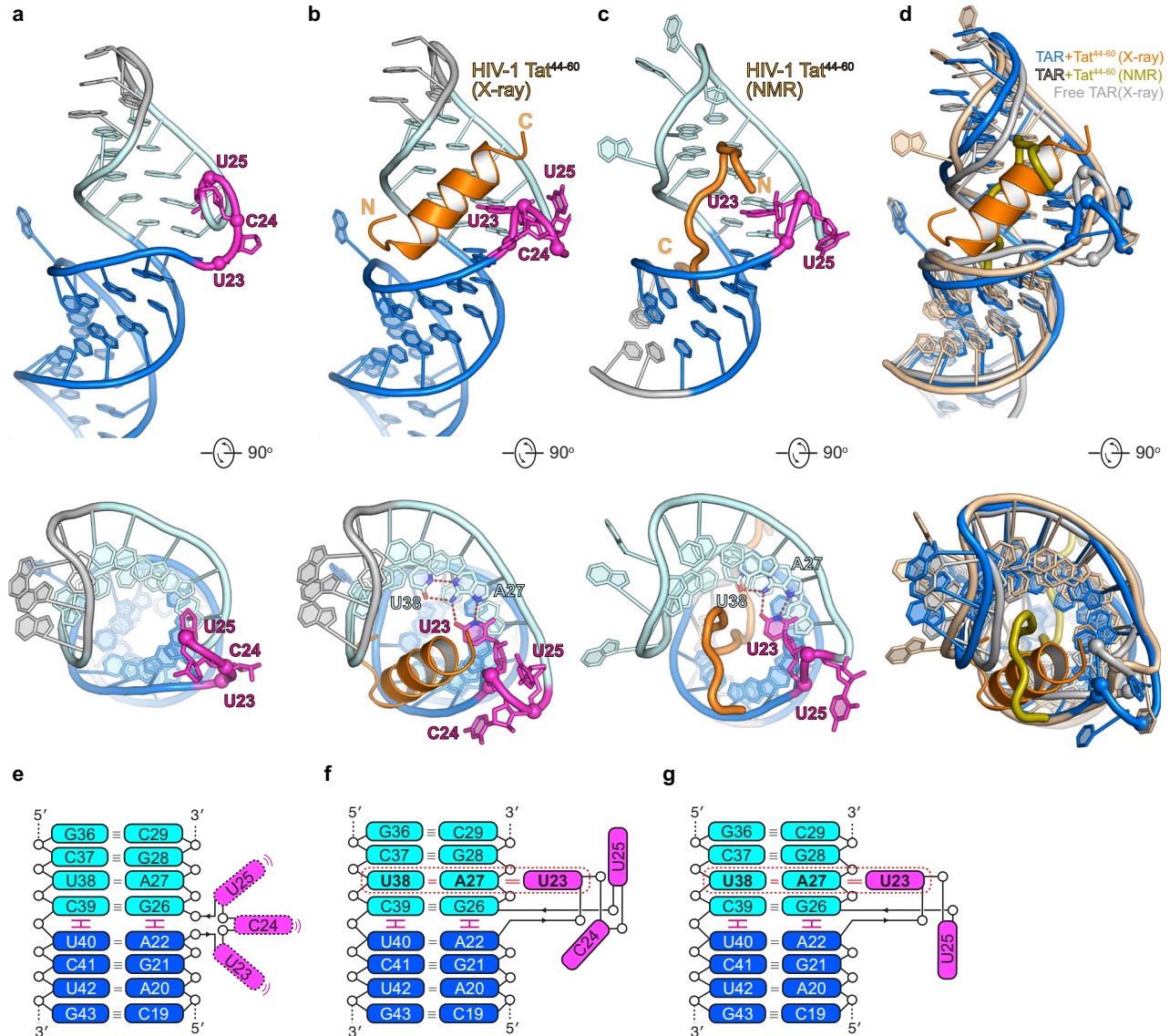

**Fig. 3 | Remodeling of HIV-1 TAR structure by HIV-1 Tat[44-60]. a, b** Front (upper) and top (lower) views of the crystal structures of full-length TAR-I in the absence (**a**) and presence of Tat[44-60] (**b**, orange). N and C indicate the termini of Tat. **c** Front (upper) and top (lower) views of the NMR structure of Tat bound to mini-TAR (PDB: 6MCE[39]). **d** Structural overlay of the free TAR-I crystal structure (gray) and co-crystal (blue and orange), and NMR (wheat and lemon) structures of the Tat-TAR complex. **e–g** Cartoon schematics of the free TAR-I crystal structure (**e**), and co-crystal (**f**), and NMR (**g**) structures of the Tat-TAR complex, highlighting the structural remodeling of the TAR UCU bulge by Tat. Red dashed boxes indicate the U23•A27-U38 base triple observed in both the co-crystal and NMR structures.

least *in crystallo* but seems unlikely in solution. Nevertheless, the specific Tat-TAR contacts observed in these crystals provide general insights into RNA interactions with IDPs and other proteins and may serve as a starting point to design novel peptide-based TAR inhibitors.

### *In crystallo* TAR-Tat helix interactions share key features with the solution NMR structure

Despite significant differences in Tat secondary structure, directionality, and conformation between the co-crystal and NMR structures, they are highly analogous in TAR conformation and local TAR-Tat contacts (Fig. 4a–f). In both cases, Tat induces a substantial conformational change which positions U23 in the major groove of the A27-U38 pair, forming a base triple (Figs. 3e–g, 4b, e). Stabilization of this base triple is central for Tat retention on the RNA, as both U23C and U23A substitutions abrogated Tat interaction (Fig. 4h, Supplementary Fig. 6 and Supplementary Table 2), in agreement with previous reports[46]. Contrasting the essential function of U23, the neighboring C24, and U25 in the bulge play ancillary roles. Any

nucleobase at position 24 sufficiently supports Tat binding, consistent with its solvent exposure and cation-π stacking with R55 of Tat in the co-crystal structure (Fig. 4a). U25 also flips out and makes no contacts to Tat. As a result, U25 can be deleted outright which slightly improved Tat binding. This finding is consistent with several NMR analyses where the trinucleotide UCU bulge was reduced to a UU dinucleotide, which is also the natural configuration in HIV-2 TAR[39].

In both the co-crystal and NMR structures, the central U23•A27-U38 base triple is sandwiched and stabilized by two G-C pairs (G26-C39 and G28-C37), as well as by two Tat arginine side chains (R52 and R56/R49) that are co-planar with the flanking G-C pairs (Fig. 4b, e). The guanidinium groups of R52 and R56/R49 not only form hydrogen bonds with the Hoogsteen edges of G26 and G28 but also stack with the base triple via cation-π and van der Waals interactions. Essentially, these two flat guanidiniums mimic third nucleobases in the major groove (like U23), forming two pseudo-base triples (R52•G26-C39 and R56/R49•G28-C37) which sandwich the central bona fide U23•A27-U38 base triple (Fig. 4b, c, e, f, triangles). Consistent with the pseudo-base

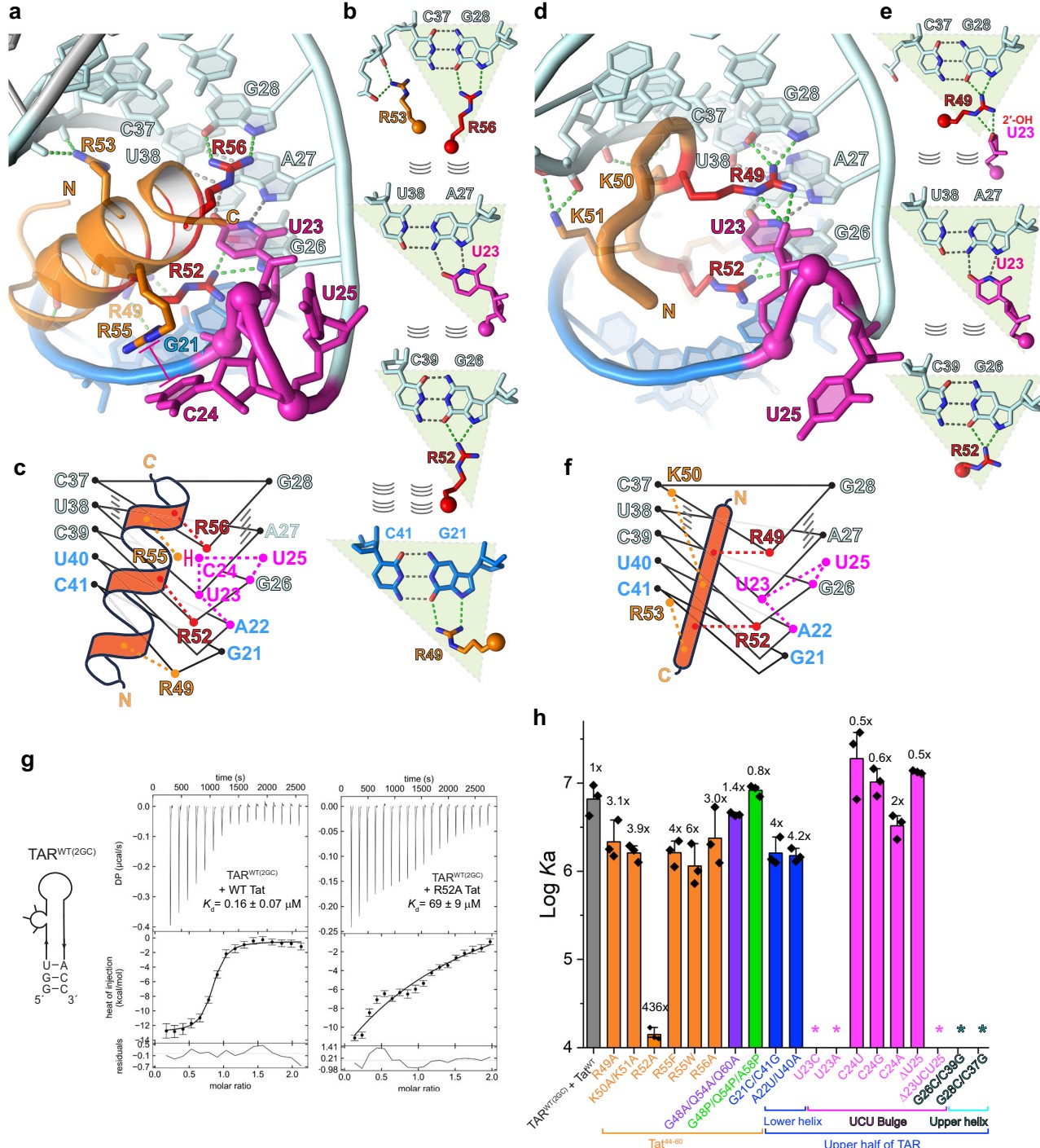

**Fig. 4 | Comparative and mutational analyses of the TAR-Tat interfaces observed in the co-crystal and NMR structures. a, d** Detailed TAR-Tat interfaces observed in the co-crystal (**a**) and NMR (**d**) structures. **c, f** Cartoon schematics of the structures in (**a**) and (**d**), respectively. Stacked triangles indicate layers of base pairs or base triples, where the vertices represent nucleobases or arginines.
**b, e** Proposed hydrogen bonding patterns within the individual base planes in (**a**) and (**d**), respectively. **g** Representative isothermal titration calorimetry (ITC)

isotherms of TAR binding to WT (left) or R52A (right) Tat[44–60]. Constructs names and $K_d$ values (mean ± s.d., $n = 3$ biologically independent samples) are indicated. **h** Summary of ITC-derived association constants ($K_a$) of TAR-Tat binding by WT and variant Tat[44–60] (left) and TAR (right). Data are mean ± s.d. of $n = 3$ biologically independent samples. *: no significant binding detected. Source data are provided in this paper.

triple formation, swapping either flanking G-C pair with an isosteric C-G pair abolished Tat interaction (Fig. 4h). This finding highlights the specificity and importance of arginine fork interactions with purine Hoogsteen edges. Both structures place R52 at the center of the interface making the same contacts, consistent with its extreme conservation. Unlike the NMR structure, the crystal structure places R49 of

Tat in direct contact with G21, making yet another arginine fork interaction with the Hoogsteen edges similar to R52 and R56 (Fig. 4b, c). Strikingly, while the flanking R49A and R56A substitutions each had only ~3-fold effects on binding, the central R52A reduced binding by more than 400 fold (Fig. 4g, h), in agreement with its in vivo importance for Tat-mediated HIV trans-activation[55,58].

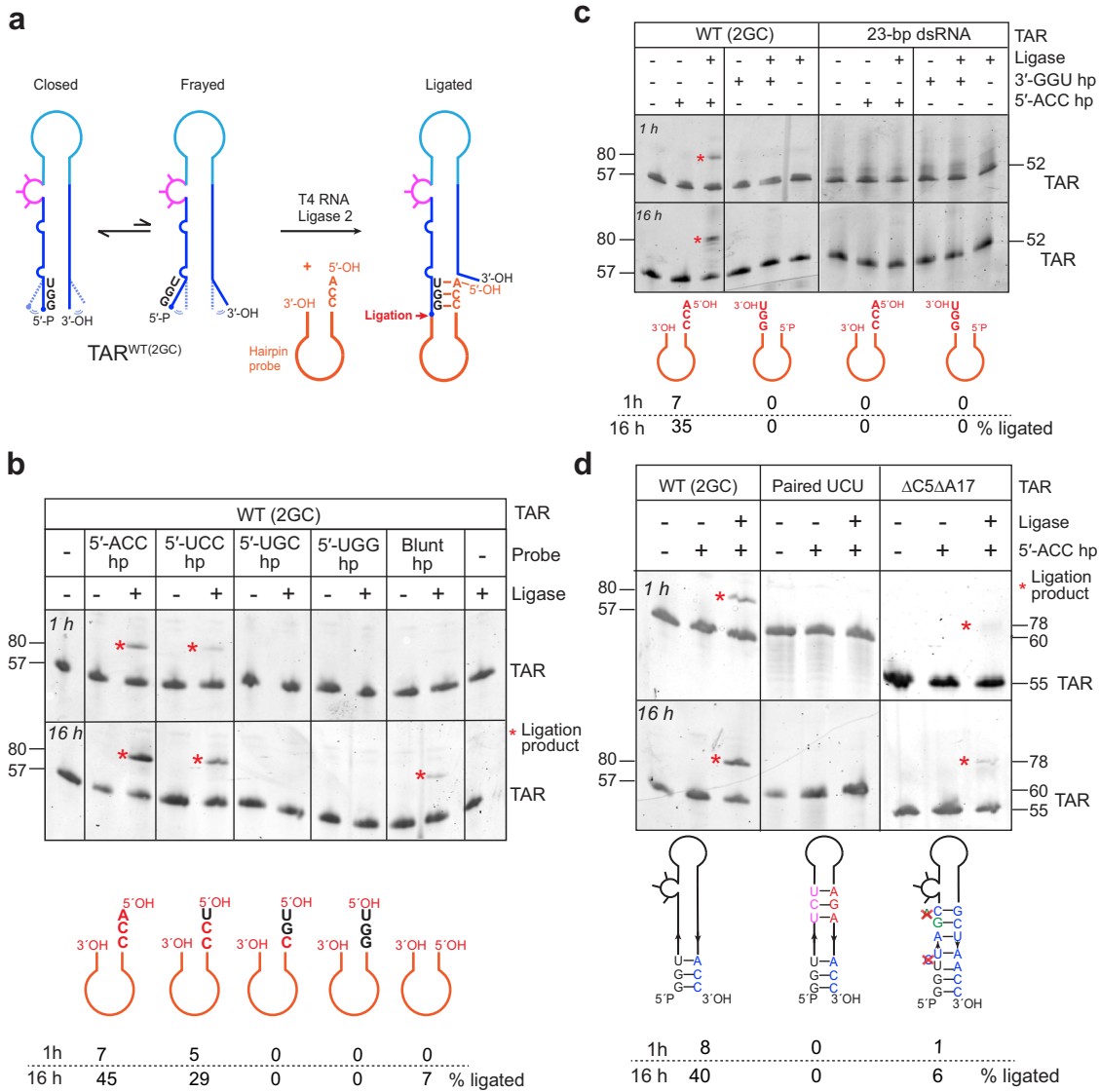

**Fig. 5 | Probing TAR termini dynamics using RNA Overhang Analysis by Ligation of Hairpin Oligonucleotides (ROALHO). a** Cartoon schematic of ROALHO. **b** Urea-PAGE analysis of ligation reactions between TAR$^{2GC}$ (WT) and five different hairpin probes. Hairpin sequences and ligation efficiencies are indicated below the gel. Independent experiments were performed twice yielding similar results. **c** Urea-PAGE analysis of ligation reactions between either TAR$^{2GC}$ (WT, left) or a 23-bp TAR-derived dsRNA (right, Supplementary Fig. 9a) and hairpin probes that bear either a 5′-ACC or 3′-GGU overhang. hp: hairpin. Experiment was performed once. **d** Urea-PAGE analysis of ligation reactions between TAR$^{2GC}$ (WT), TAR with a paired UCU bulge, or TAR$^{\Delta C5\Delta A17}$ and the 5′-ACC hairpin probe. Independent experiments were performed twice yielding similar results. The RNA lengths are indicated in nucleotides. Source data are provided in this paper.

Overall, the alpha-helical, *in crystallo* conformation of Tat binds at a similar location as the linear peptide observed in solution by NMR spectroscopy. Interestingly, Tat binding in either direction drove the formation of the essential U23•A27-U38 base triple. Tat projects the guanidinium groups of three arginines, R49, R52, and R56, to recognize the Hoogsteen edges of three guanines, G21, G26, and G28, respectively, with R52 making a central, indispensable contact. The unusual ability of Tat peptide to pivot about its R52 anchor to engage TAR in two distinct conformations and main-chain directions suggest a remarkable versatility of arginine-rich motifs in RNA recognition. Nonetheless, our helicity modulation data suggest that the alpha-helical conformer of Tat is dispensable for TAR binding in solution.

## RNA Overhang Analysis by Ligation of Hairpin Oligonucleotides (ROALHO) corroborates TAR lower stem metastability
Our RNA melting and TAR-cTAR annealing analyses showed that single-nt modifications to the WT TAR termini modulate overall TAR stability and its annealing with cTAR (Supplementary Figs. 2 & 3). We reason that this is likely due to an inherently metastable lower stem that could transiently fray open to facilitate annealing with cTAR. This metastability is further supported by the crystallographically observed conformational excursions of C5 (Fig. 2a–c, g–i). Next, we sought to further probe the TAR lower stem structural dynamics using another, orthogonal method. We reasoned that an exogenous hairpin oligo bearing a complementary overhang to the TAR terminal region may be able to base pair with a transiently open TAR termini allowing enzymatic ligation by T4 RNA Ligase 2 (Rnl2, Fig. 5a). Rnl2 seals nicks between juxtaposed 3′-OH and 5′-monophosphate groups in dsRNA. Similar ligation-based approaches have been previously used to probe phosphorylation states of RNA 5′ ends or to assess tRNA 3′ end aminoacylation[59,60]. Indeed, we observed that Rnl2 catalyzed robust ligation (~ 45% at 16 h) between WT TAR and a hairpin probe bearing a complementary 5′-ACC overhang (Fig. 5b, asterisk). Next, we assessed the RNA end requirements, sensitivity, and specificity of this assay

which we term RNA Overhang Analysis by Ligation of Hairpin Oligonucleotides (ROALHO, Methods, Supplementary Fig. 9). We find that ligation requires at least two complementary base pairs between the hairpin overhang and TAR, and that the TAR 5′ end bearing a monophosphate is more readily ligated than its 3′ end bearing a 3′-OH with their complementary hairpin probes that carry the monophosphates (Fig. 5b, c and Supplementary Fig. 9b).

We then used ROALHO to examine how various structural elements of TAR contribute to the lower stem malleability and transient fraying of its termini. Remarkably, a perfectly paired 23-bp TAR-like hairpin without any bulges exhibited no ligation to either the 5′-ACC or 3′-GGU hairpins, suggesting that at least one bulge is required to confer metastability in the TAR lower stem (Fig. 5c). To dissect the relative contributions of the two adjacent single-nt bulges (C5 and A17) versus the distant trinucleotide UCU bulge, we removed them separately. Removing the C5 and A17 bulges reduced ligation by ~ 7 fold while pairing up the UCU bulge by inserting a complementary AGA trinucleotide opposite the bulge abolished ligation (Fig. 5d). Together, these findings confirm the metastability of the TAR lower stem, and further suggest that this conformational flexibility is conferred by the three bulges. Interestingly, the larger UCU bulge, located 18-bp away from the termini, exerts a larger impact than the much closer single-nt bulges. This further corroborates the observed contiguous base stacking throughout the entire TAR hairpin and demonstrates that structural effects of RNA bulges can propagate long distances along dsRNAs to affect the termini. As the dsRNA grows in length, its overall stacking strength and stability increases accumulatively, thus allowing distant structural features such as bulges to modulate the thermodynamic behavior of the termini. Similar long-range effects were previously observed in the T-box riboswitch-tRNA complex, where the overall stability of a 33-bp-long "central spine" assembled from four coaxially arranged dsRNA segments determines the regulatory outcome of the riboswitch[61,62]. These data also suggest that ROALHO may provide a general strategy to probe for transient openings and conformational excursions of nucleotides near RNA termini, by sequence-specific strand exchange and product enrichment by ligation.

## Multi-probe, time-resolved fluorescence spectroscopy reveals differential protein remodeling of TAR structure

To probe how individual regions of TAR coordinately change conformations during encounters with protein factors, we incorporated 2-aminopurine (2AP) individually at four different positions along the TAR hairpin, at G3, A17, A20, or C24 (Fig. 6a). The quantum yield and lifetime of 2AP are exquisitely sensitive to its local chemical environment, and is primarily quenched by stacking interactions with its nearest neighbor nucleobases[63–66]. 2AP lifetime analysis has the precision and resolving power to distinguish between one-sided stacking from two-sided stacking[61,67]. Notably, due to instability and multiple pairing configurations of the 2AP-C mispair[68], we introduced a C57U substitution opposite 2AP at the G3 position to ensure Watson-Crick pairing at this location (Fig. 6c, italic). Although this change was necessary, it may have slightly altered the local structural dynamics.

We first measured 2AP lifetimes at various probe locations within the free TAR. 2AP probes located at G3 (dsRNA near termini), A17 (single-nt bulge), A20 (dsRNA mid-section), and C24 (UCU bulge) positions exhibited a wide range of average lifetimes of ~ 2.6, 7.3, 1.4, and 4.4 ns, respectively (Fig. 6a, b, Supplementary Fig. 10a–d and Supplementary Table 5). Importantly, 8 M urea nearly equalized all these lifetimes, to 2.4, 3.5, 2.2, and 2.6 ns, respectively. This suggests that the substantial differences in 2AP lifetimes under native conditions primarily reflect differences in structural but not sequence contexts, which are congruent with our crystal structures and affirm the lower stem metastability. Despite similar dsRNA contexts, 2AP3 displayed a lifetime (2.6 ns) nearly double that of 2AP20 (1.4 ns), indicative of a loosely paired terminal region prone to fraying or strand

exchange. Both 2AP17 and 2AP24, expected in bulged-out conformations, indeed showed much longer lifetimes. Interestingly, 2AP17 is stably bulged out and exhibited an unusually long lifetime of 7.7 ns, approaching the ~ 10 ns of free 2AP in solution. By contrast, the shorter lifetime (4.4 ns) of 2AP24 is consistent with occasional stacking with the neighboring U23 facilitated by divalent cations (Supplementary Fig. 4g, h).

With a baseline established for the free TAR, we assessed the impact of NC, Tat, and dsRBMs of PKR, which exhibited divergent patterns of effects consistent with their distinct binding sites and modes (Fig. 6a). We generally associate lower lifetimes with enhanced ability to interact (e.g., stack) with nearest neighbors[69]. In protein binding studies, we assign this enhanced quenching interaction to a disruption in the structure by the binding partner that allows the 2AP to revert to a simpler stacked conformation. This does not preclude other quenching interactions such as those from the solvent. Conversely, we interpret protein binding-induced lifetime increases to come from either stabilization of nonquenched conformers (e.g., flipped out), or destabilization of 2AP-quenching local structures.

NC destabilizes RNA secondary structures by capturing single-stranded, accessible guanosines[7,70]. Consistent with this, we find that NC substantially increased the lifetimes of 2AP3, 2AP20, and 2AP24 but decreased the lifetime of 2AP17, as expected for an order-to-disorder transition from a dsRNA hairpin to an unstructured ssRNA (Fig. 6c–f). In particular, NC denaturation of TAR would allow the otherwise stably flipped, unquenched 2AP17 to stack with its neighboring nucleobases in a ssRNA context, resulting in a reduction of its lifetime from 7.3 to 5.3 ns. Contrasting NC, Tat generally stabilizes the TAR structure. Tat binding to TAR progressively reduced 2AP3 lifetime from 2.6 to 1.8 ns (Fig. 6c). This is likely mediated by long-range propagation of UCU bulge stabilization by Tat to the TAR termini, in line with our finding that pairing up the UCU bulge blocked hairpin ligation to the termini (Fig. 5d). Tat binding also modestly reduced 2AP17 lifetime from 7.2 to 6.0 ns (Fig. 6d). This effect may be due to Tat binding the UCU bulge proximal to 2AP17, or excess Tat transiently and nonspecifically engaging and quenching the solvent-exposed 2AP17. Tat strongly quenched 2AP20, reducing the already short lifetime of 1.4 ns to 0.15 ns (Fig. 6e). This is presumably due to Tat binding near 2AP20 and locally stabilizing the duplex. Tat increased the lifetime of 2AP24 from 4.4 to 6.2 ns (Fig. 6f), consistent with its stabilization of the U23 triple and concomitant stable extrahelical flipping of C24. The binding of the tandem dsRBMs of PKR to TAR produced yet another distinct pattern of quenching. Like Tat, the dsRBMs substantially reduced the lifetime of 2AP3 (from 2.6 to 1.0 ns, Fig. 6c, black lines), consistent with its direct binding to the TAR lower stem and immobilization of the otherwise flexible termini. By contrast, 2AP17 was largely unaffected by dsRBMs binding (Fig. 6d), consistent with its already stable extrahelical conformation and A17's minimal contribution to PKR binding or inhibition. Unlike 2AP3 and 2AP17, dsRBMs increased the lifetime of 2AP20 from 1.4 to 2.5 ns (Fig. 6e), indicative of the long-range effect of its binding. Interestingly, dsRBMs binding increased the lifetime of 2AP24 from 4.4 to 6.1 ns, in a pattern very similar to Tat (Fig. 6f). One interpretation of the observation is that tightening the TAR lower stem by PKR binding may help structure the UCU bulge, predisposing it toward the Tat-bound conformation featuring the U23 triple. NC seems to initially chaperone a similar structuring of the UCU bulge, but when in greater excess, eventually destabilizes the global structure and thus reduces the 2AP24 lifetime by allowing intra-strand stacking with its neighboring nucleobases in a single-stranded context.

## Discussion

The main findings of this study are: (1) the full-length HIV-1 TAR RNA structure features a dynamic UCU bulge conformationally coupled to a metastable, fraying-prone lower stem. (2) TAR is a robust inhibitor of PKR and uses its lower stem to capture its dsRBMs. (3) under certain

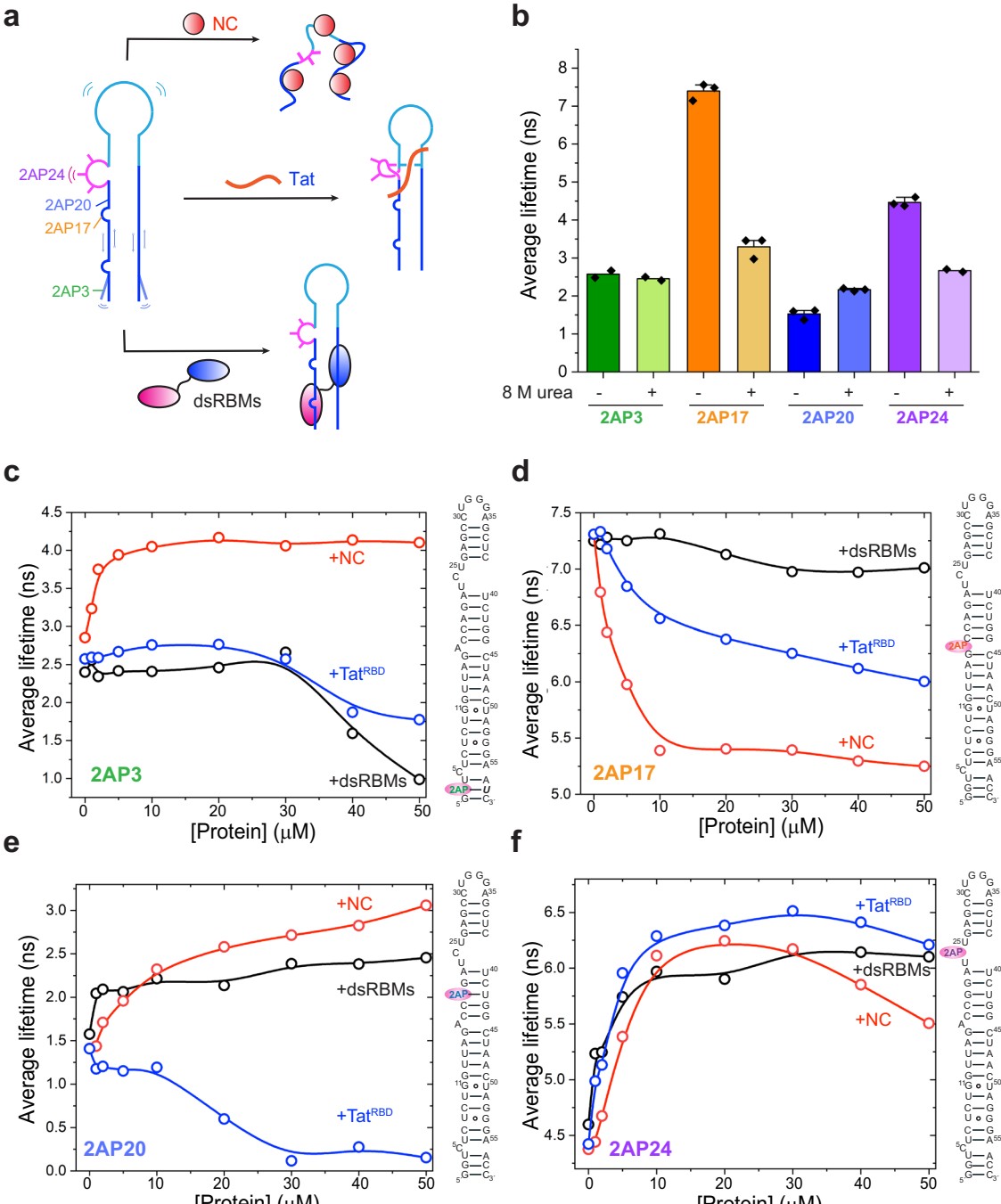

**Fig. 6 | Protein-induced structural remodeling of TAR monitored by 2AP time-resolved fluorescence spectroscopy. a** Cartoon schematics of TAR conformational changes upon binding of NC, Tat, or the dsRBMs of PKR. **b** Average fluorescence lifetimes of 2-aminopurine (2AP) at positions 3 (green), 17 (orange), 20 (blue), or 24 (purple) of free TAR$^{2GC}$, under native conditions (solid colors) or in 8 M Urea (pastel colors). Data are mean ± s.d. of $n$ = 3 biologically independent samples for 2AP17, 2AP20, and 2AP24 without Urea, or mean of $n$ = 2 biologically

independent samples for 2AP3 and 2AP with Urea. **c**–**f** Average fluorescence lifetimes of 2AP at position 3 (**c**), 17 (**d**), 20 (**e**), and 24 (**f**) of TAR$^{2GC}$ in the presence of increasing concentrations of NC (red), dsRBMs of PKR (black), or Tat$^{RBD}$ (Tat$^{44-60}$, blue). The position of 2AP is indicated by a red oval in the secondary structures. A C57U substitution (bold, italic) was introduced to ensure its stable Watson-Crick pairing with 2AP at position 3 (**c**). Source data are provided in this paper.

conditions, the Tat peptide could engage TAR in its alpha-helical form making similar interactions to the UCU bulge as the NMR structure. (4) the dynamic, multi-bulge TAR structure is poised for local and global structural remodeling by proteins including Tat, NC, and PKR.

An expanding cohort of host and viral RNAs of various lengths, topologies, and structures act on PKR to elicit a wide spectrum of critical downstream effects including stress response, apoptosis, and inflammation[18]. This trend was further accentuated by the recent

identification of endogenous circular and mitochondrial RNAs as important PKR regulators[71,72] However, the molecular mechanisms that correlate the structural features of these RNAs with their effects on PKR have remained elusive[18]. Our findings show that TAR directly targets PKR using its lower stem, employs a strategy similar to VA-I, and achieves robust inhibition. Previous studies revealed that TAR dimers and co-purified dsRNA elements activate PKR, which can further straighten bent and bulged dsRNAs to induce its own activation[32,33].

Given the flexibilities of several regions of TAR and its tendency to dimerize via strand exchange, different solute, folding, or purification conditions may have produced distinct TAR conformations or oligomeric states, which in turn elicited divergent modulatory activities on PKR. The examples of TAR, VA-I[44], and circular RNAs[72] converge on the notion that a 24–28 bp imperfectly paired dsRNA segment is an effective inhibitor of the kinase and is likely used by other PKR-regulatory RNAs.

A comparative analysis of the crystal structures of full-length TAR revealed that the C5 bulge can sample multiple locations, conformations, and contacts. These findings are corroborated and expanded by ROALHO and 2AP lifetime analyses, two orthogonal, general methods that probe RNA structural dynamics in solution. Both assays further uncovered strong conformational coupling between the upper and lower halves of TAR, and suggest a bidirectional linkage between the UCU bulge and the termini via coaxial stacking. Notably, previous enzymatic and chemical probing experiments supported TAR lower stem metastability and terminal fraying, which was proposed to enable long-range interactions with other elements of the leader[32,73]. Besides TAR, several other RNA systems exemplify how dynamic behaviors of dsRNA elements can contribute to function and regulation. The precursor of microRNA-21 contains a short, dynamic apical stem that spontaneously shifts its pairing register to modulate Dicer cleavage and microRNA maturation[74]. Interestingly, this bistable stem shares several key features with the TAR lower stem, including the use of two less-stable G•U wobble pairs in the ground state, single-nt bulges that help drive the shift, and immediately adjacent helices that power the shifts through coaxial stacking[74]. Similarly, G•U wobble pairs enriched in the acceptor stem of the tRNA-like menRNA facilitate a spontaneous register shift and conformational isomerization to drive double CCA addition and rapid degradation[65]. Besides naturally occurring RNAs, in vitro selection has created artificial "slipping helices", which can slide and shift their pairing registers depending on ligand binding to an adjacent aptamer domain[75]. This allowed them to serve as effective allosteric communication modules between RNA domains.

Our data suggest that the malleable structure of TAR is poised for binding and structural remodeling by external factors. Although the U23•A27-U38 triple is not observed in free TAR structures, it appears predisposed to form and thus represents a hidden, low-energy state that is rapidly populated when arginine-containing factors approach the bulge. These include argininamide, Tat peptide in both directions, a lab-evolved RNA recognition motif (RRM) termed TBP6.7, a β-hairpin macrocyclic peptide termed JB181, and others that carry guanidinium groups[38,76]. For comparison, $Mg^{2+}$ and $Ca^{2+}$ also remodel the bulge but are insufficient to drive stable triple formation. Topologically or conformationally constrained peptides such as TBP6.7 and JB181 are among an emerging class of promising therapeutics with significantly larger target-binding interfaces than small molecules. Although our Tat helix-TAR structure seems to require the crystal lattice for stabilization, it demonstrates the structural feasibility for a conformationally constrained Tat helix to fit in the TAR major groove, induce the U23 triple formation, and make highly similar contacts. It might also shed light on the mechanisms of Tat-binding RNA aptamers[77], and hint at the existence of natural RNA substrates that Tat may bind in its alpha-helical form[78].

Despite the clear promise of targeting TAR with small molecules or peptides, none have yet entered pre-clinical trials to date, presumably due to their lower affinity and specificity[4,79]. The polymorphic, dynamic structures of TAR may have exasperated this difficulty, but also provide opportunities to target the transient, high-energy states that may provide the much-needed specificity. This study provides a more complete picture of the TAR structure and its dynamics and reveals extensive conformational coupling between the upper and lower stems. The metastable lower stem, containing two G•U wobble pairs and two single-nt bulges, is reminiscent of the "P1" helix of most

riboswitches that dictates overall RNA conformation and gates gene expression[80]. The dynamic behavior of the TAR lower stem not only impacts the conformational dynamics of the UCU bulge of the upper stem, but may also modulate the coaxial stacking interaction with its immediately adjacent polyA stem, and in turn, control the conformation of the entire 5′ UTR and influence the decision between virion packaging and translation[80,81]. Further, the TAR constructs used in this study lack the 5′ cap, which is covalently linked to and can stack with the immediately adjacent 5′G or a blunt end base pair involving the 5′ G[13,14]. It will be of interest to further explore how the presence of the 5′ cap, followed by one, two, or three Gs, may modulate the TAR structural dynamics. Taken together, our combined structural and biophysical analyses of the full-length HIV-1 TAR RNA may inform the design of next-generation compounds or peptides that target TAR or the larger 5′ UTR region.

## Methods
### RNA preparation
All RNAs used for crystallization and biochemical studies were transcribed in vitro using T7 RNA polymerase as described previously[82,83]. Specifically, PCR products bearing two consecutive 2′-O-methyl modifications on the 5′-end of the template strand were used as templates for transcription in the presence of 5 mM of each nucleotide triphosphate (ATP, CTP, UTP, and GTP). RNAs were purified on an 8 M urea denaturing preparative 10% polyacrylamide (29:1 acrylamide:bisacrylamide) gel and electroeluted. Extracted RNAs were washed once with 1 M KCl, desalted by washing three times with DEPC-treated water, and stored at – 20 °C.

RNAs used for ligation experiments were prepared by in vitro transcription as described above with the exception of the addition of 20 mM GMP in the presence of 1 mM GTP. This ratio of nucleotides allows GMP priming to introduce a 5′ monophosphate on the desired RNA product. RNA with a 5′-OH were prepared by in vitro transcription with a hammerhead ribozyme upstream of the desired product. The 23-bp RNA duplex and DNA oligos were chemically synthesized by Integrated DNA Technologies (IDT). All RNAs were refolded prior to use by heating at 95 °C for 2 min, cooling on ice, and addition of 2 mM $MgCl_2$. All RNA mutants were generated by site-directed mutagenesis using the QuikChange Lightning kit (Agilent). Periodate oxidation of RNA was used for 6-carboxyfluorescein (FAM) labeling at the 3′-end. 10 μM of RNA was incubated with 0.1 M sodium meta-periodate and 0.1 M sodium acetate pH 5.2 for 90 min at 25 °C in a total volume of 200 μL. The reaction was quenched with the addition of 0.25 mM KCl and incubated on ice for 10 min leading to the formation of a white precipitate. The supernatant was recovered and desalted using a MicroSpin™ G25 column (Cytiva) and incubated with 2 mM FAM-hydrazide in 0.1 M sodium acetate pH 5.2 for 16 h at 25 °C. The RNA was recovered by ethanol precipitation, washed once with 70% ethanol, and buffer exchanged in DEPC-treated water using a MicroSpin™ G25 column. The labeling efficiency was determined by measuring the concentration of FAM on a NanoDrop spectrophotometer using an extinction coefficient of 75000 $M^{-1}cm^{-1}$ at 495 nm. 2-aminopurine (2AP) containing RNAs were obtained from Horizon Discovery or IDT.

### Crystallization and structure determination
To facilitate the crystallization of full-length TAR, the apical loop was partially substituted by a GAAA tetraloop known to promote crystal contact through A-minor interactions[82,83]. All crystal structures were obtained by vapor diffusion at 20 °C by mixing 1:1 - 7 mg/mL of RNA in 25 mM Tris-HCl pH 7.5, 150 mM NaCl, 2 mM $MgCl_2$, and 1 mM spermine with reservoir solutions defined below. The full-length TAR with G16-A17 swapped to A16-G17, and an additional A at the 3′ end (TAR$^{G16A/A17G-GAAA-ins59A}$, TAR-II) was crystallized by mixing the RNA solution 1:1 with a reservoir solution of 50 mM sodium cacodylate pH 6.5, 80 mM NaCl, 12 mM spermine and 30% v/v 2-methyl-2,4-pentanediol

(MPD). The crystals were cryoprotected in a similar reservoir condition by increasing the MPD content to 60% v/v and vitrified at 100 K. For single-wavelength anomalous (SAD) data, the crystals were soaked for 1 day at 20 °C with 5 mM Ir (III) hexaammine prior to vitrification. The structure was solved by MR-SAD with an initial molecular replacement (MR) solution found using an 8 bp ideal dsRNA fragment using PHASER[84]. The initial MR solution produced an overall TFZ (translation function Z-score) of 11 and a LLG (log-likelihood-gain) of 138.4. Nine datasets were analyzed and combined using the hierarchical agglomerative clustering program BLEND[85] producing an initial figure of merit (FOM) of 0.57.

TAR$^{GAAA}$ (TAR-I) crystallized in 12 mM NaCl, 80 mM KCl, 50 mM sodium cacodylate pH 5.5, 2 mM Cobalt (III) hexaammine and 45% v/v MPD. The crystals were cryoprotected by increasing the MPD content to 60% v/v prior to vitrification at 100 K. For SAD data, the crystals were soaked for 3 days at 20 °C with 5 mM Ir (III) hexaammine prior to vitrification. The structure was solved by MR-SAD with an initial MR solution obtained using the structure of TAR-II as a starting model with PHASER. The initial MR solution produced an overall TFZ of 7.9 and an LLG of 353. A single dataset on a single crystal exhibited strong anomalous signal producing an initial FOM of 0.48. The co-crystal structure of TAR-I bound to Tat$^{44-60}$ was obtained by soaking overnight at 20 °C the TAR-I crystals in a reservoir solution supplemented with 0.5 mM Tat. The Tat$^{44-60}$ peptide was obtained from GenScript and is of the sequence: GISYGRKKRRQRRRAHQ. The structure of TAR-II containing Ca$^{2+}$ was obtained by soaking the TAR-II crystals with 100 mM CaCl$_2$ for 2 hours at 20 °C. All diffraction data were collected at SER-CAT beamline 22-ID at the Advanced Photon Source (APS). X-ray diffraction data were indexed, integrated, and scaled using XDS. Ir-SAD data were collected at a wavelength of 1.1 Å. Iterative rounds of model building was performed in Coot[86] and refined using Phenix.Refine. The models were corrected by ERRASER[87] and further refined in Phenix.

### Protein expression and purification

Human PKR was co-expressed with λ-phosphatase in *E. coli* BL21(DE3) RIL cells and induced with 0.5 mM isopropyl 1-thio-β-D-galactopyranoside (IPTG) at O.D.$_{600}$ ~ 0.6 for 2 h at 30 °C[44]. Cells were lysed with a microfluidizer in 25 mM Tris-HCl pH 7.5, 500 mM NaCl, 15 mM imidazole, 10% glycerol, and 10 mM β-mercaptoethanol supplemented with the SIGMAFAST$^{TM}$ Protease inhibitor cocktail. The clarified supernatant was loaded on a HisTrap Ni$^{2+}$ column on an AKTA Pure chromatography system and eluted with 250 mM imidazole. The eluted PKR was further purified using a heparin column with a salt gradient from 50 mM to 1 M NaCl and a subsequent Superdex 200 increase 10/300 GL column equilibrated in 25 mM Tris-HCl pH 7.5, 200 mM NaCl, 15% glycerol, 1 mM DTT and 2 mM MgCl$_2$. The dsRBMs construct was co-expressed with λ-phosphatase in *E. coli* BL21(DE3)RIL and induced with 0.5 mM IPTG at O.D.$_{600}$ ~ 0.6 for 3 hours at 37 °C. The dsRBMs were purified using the same protocol as for unphosphorylated PKR with an additional step of tag cleavage prior to the heparin column. The tag was cleaved by adding the TEV protease at a 1:300 TEV:dsRBMs ratio and dialysis for 16 h at 4 °C in 25 mM Tris-HCl pH 7.5, 200 mM NaCl, and 1 mM DTT. HIV-1 nucleocapsid protein (NC) was cloned in a pD441-HMBP vector (ATUM). NC was expressed in *E. coli* BL21(DE3)RIL cells grown in LB medium supplemented with 0.1 mM ZnCl$_2$. The cells were induced at O.D.$_{600}$ ~ 0.6 with 1 mM IPTG for 3 h at 37 °C. The protein was purified using the same protocol as for PKR except that the heparin step was omitted. The maltose binding protein (MBP) fusion tag was cleaved by adding PreScission protease at a 1:300 PreScission:NC ratio and dialysis for 16 h at 4 °C in 25 mM Tris-HCl pH 7.5, 200 mM NaCl, and 1 mM DTT.

### PKR kinase assay

All PKR kinase activity assays were performed with 100 nM of unphosphorylated PKR and varying amounts of RNAs in 25 mM Tris-HCl pH 7.5, 50 mM NaCl, 50 mM KCl, 1 mM DTT, and 2 mM MgCl$_2$. RNAs were incubated with PKR for 10 min at 25 °C, and the reactions were started by adding 5 μM ATP and incubating the mixture at 30 °C for 15 min. Phosphorylation of PKR was quantified by measuring the formation of ADP using the ADP-Glo$^{TM}$ kinase assay (Promega). The reactions were quenched by adding 20 μL of ADP-Glo$^{TM}$ reagent to a 20 μL reaction and incubated for 40 min at 25 °C prior to the addition of 10 μL of kinase detection reagent for 30 min. The luminescence signal was recorded on a BMG CLARIOstar Plus plate reader. To assess PKR inhibition, varying amounts of RNAs were pre-incubated with PKR for 10 min at 25 °C followed by the addition of 10 nM of 79-bp dsRNA and 5 μM ATP and further incubation at 30 °C for 15 min. The following equation was used to determine the IC$_{50}$ in Origin version 2018:

$$y = A1 + \frac{(B1 - A1)}{1 + 10^{(\log(IC50)-X)p}} \tag{1}$$

A1 and B1 are the maximum and minimum activity, respectively. p is the Hill slope coefficient that was fixed to −1 while other parameters were allowed to float during the fit.

### Binding measurements by fluorescence polarization (FP)

5 nM of RNAs labeled with fluorescein were titrated with increasing amounts of dsRBMs or PKR (up to 8 μM) in 25 mM Tris-HCl pH 7.5, 50 mM NaCl, 50 mM KCl, and 2 mM MgCl$_2$ in a 96-well plate at 21 °C. FP values were measured in triplicates using a BMG CLARIOstar Plus microplate reader with excitation at 482 nm, emission at 530–540 nm, and LP (long pass) 504 nm dichroic filter setting. Changes in FP (ΔFP) as a function of protein concentrations were fit with the following equation to determine the apparent dissociation constant $K_d$ in Origin Version 2018:

$$y = \frac{\Delta FP_{max} * x}{K_d + x} \tag{2}$$

### Competition experiments by fluorescence polarization

5 nM of labeled 79-bp dsRNA was mixed with 100 nM PKR in 25 mM Tris-HCl pH 7.5, 50 mM NaCl, 50 mM KCl, and 2 mM MgCl$_2$ and incubated for 10 min at 21 °C. Then increasing amounts of unlabeled RNAs were added to compete for PKR binding in a 96-well plate. FP values were recorded using the same settings as for the binding measurements above. Changes in FP (ΔFP) as a function of competitor concentrations were fit to the following equation to determine the apparent half-maximal inhibitory concentration (IC$_{50}$):

$$y = \Delta FP_{final} + \frac{\left(\Delta FP_{initial} - \Delta FP_{final}\right)}{\left(1 + 10^{(\log(x) - \log(IC50))}\right)} \tag{3}$$

### Isothermal titration calorimetry (ITC)

The Tat$^{44-60}$ peptide was dissolved in a buffer consisting of 25 mM Tris-HCl (pH 7.5), 100 mM NaCl, and 1 mM MgCl$_2$, and TAR constructs were extensively exchanged into the same buffer using Amicon Ultra Filter concentrators (Millipore). All ITC measurements were performed at 25 °C using a MicroCal iTC200 microcalorimeter (GE healthcare). 10 μM of RNA was used in the cell and titrated with 100 μM of Tat$^{44-60}$ from the syringe. The raw ITC data were integrated using NITPIC and fit with SEDPHAT[88] to obtain the dissociation constants and thermodynamic parameters.

### Size-exclusion chromatography coupled to multi-angle light scattering (SEC-MALS)

To assess binding stoichiometries of PKR or its dsRBMs with TAR, 40 μM of TAR and 45 μM of protein were incubated for 10 min at room

temperature in a buffer consisting of 25 mM Tris-HCl pH 7.5, 50 mM NaCl, 50 mM KCl, and 2 mM $MgCl_2$, prior to injection onto a Superdex 200 Increase column on an Agilent HPLC system equilibrated in 25 mM Tris-HCl pH 7.5, 50 mM NaCl, 50 mM KCl and 2 mM $MgCl_2$. The HPLC system was coupled to a DAWN HELEOS II detector equipped with a quasi-elastic light scattering module and an Optilab T-rEX refractometer (Wyatt Technology). Data were analyzed using the ASTRA 7.3 software (Wyatt Technology Europe).

### Differential scanning calorimetry (DSC)

DSC experiments were performed with 20 μM of RNA in 25 mM HEPES pH 7.5, 150 mM NaCl, and 2 mM $MgCl_2$ on a Malvern/GE VP-DSC instrument. To assess the impact of $Mg^{2+}$ on the melting profiles, the experiments were repeated in 25 mM HEPES pH 7.5, 150 mM NaCl, and 0.5 mM EDTA. The DSC instrument was equilibrated overnight with buffer in both sample and reference cells. The next day, the RNA was loaded in the sample cell, and the DSC scan was recorded after a 60 min equilibration. The temperature range scanned was from 25 to 120 °C with the step size of 1 °C $min^{-1}$. DSC data were corrected for instrument baselines and normalized for scan rate and duplex concentration. Data conversion and analysis was performed with Origin software (OriginLab Corporation, Northampton, MA, USA).

### Circular dichroism (CD)

To determine the thermal stability of TAR, CD experiments were performed with 10 μM of RNA in 25 mM Tris-HCl pH 7.5, 150 mM NaCl, with or without 2 mM $MgCl_2$ on an Applied Photophysics Chirascan$^{Tm}$ Q100 spectropolarimeter. The CD spectra were recorded from 195 to 320 nm using a 1 mm pathlength cell. The temperate range scanned from 20 to 97 °C at a rate of 1 °C $min^{-1}$. The CD data were analyzed using the Global3 software (Applied Photophysics). To assess the secondary structure of Tat$^{44-60}$ in solution, CD experiments were performed with 300 μM of peptide in 25 mM Tris-HCl pH 7.5 and 25 mM NaCl at 20 °C. The CD spectra were recorded from 185 to 260 nm using a 0.1 mm path length cell. The data were analyzed in dichroweb[89] using the CDSSTR program with reference set 6 (optimized for 185–240 nm) containing soluble and denatured proteins.

### Time-resolved fluorescence spectroscopy using TCSPC (Time correlated single photon counting)

10 μM of 2AP-labeled RNA in a buffer containing 25 mM Tris-HCl pH 7.5, 150 mM NaCl, and 2 mM $MgCl_2$ was incubated with increasing amounts of protein. To measure the lifetimes of denatured TAR, the RNA was diluted 10-fold directly into 8 M urea. The fluorescence lifetimes were measured on a FluoroMax Plus Spectrofluorometer equipped with DeltaTime TCSPC (Time-Correlated Single Photon Counting) module (Horiba Scientific). The peak excitation wavelength was centered at 310 nm. The instrument response function (IRF) was recorded using a solution of Ludox colloidal silica. 2AP emission was recorded at 370 nm using a 20 mm path length until 10, 000 counts were reached. The data were analyzed in the DataStation v2.4 software (Horiba Scientific) and all 2AP fluorescence decays were fit with 3 exponentials. Confirmatory experiments with higher time resolution were made on a purpose-built TCSPC instrument using ps laser pulses and a Hamamatsu R3809 MCP-PMT to achieve IRF with full width at half maximum (FWHM) < 120 ps[90]. The mean lifetimes from both instruments were in substantial agreement, indicating that the 2AP quenched environments were not dominated by sub-ns terms.

### RNA Overhang Analysis by Ligation of Hairpin Oligonucleotides (ROALHO)

For ligation experiments, individual RNAs were refolded separately by heating at 95 °C for 2 min followed by incubation on ice with the addition of 2 mM $MgCl_2$. 10 μM of each RNA was then mixed in 25 mM Tris-HCl pH 7.5, 150 mM NaCl, and 2 mM $MgCl_2$ supplemented with 1X

of ligation buffer (#B0239S) from New England BioLabs (NEB) in a final volume of 30 μL. Ligation was initiated by adding 1 μL of T4 RNA ligase 2 (NEB), and the mixture was incubated at 25 °C for 1 h or 16 h. The reactions were quenched by mixing 1 μL of the ligation mixture with 19 μL of loading dye composed of 95% v/v formamide, 0.1% w/v SDS, 1 mM EDTA, xylene cyanol, and bromophenol blue, and analyzed on a 10% Urea-polyacrylamide (29:1 acrylamide:bisacrylamide) gel stained with SYBR$^{TM}$ Gold (ThermoFisher). Band intensities were quantified using ImageJ. All TAR RNAs used in ROALHO bear the natural hexanucleotide (CUGGGA) apical loop. To verify the chemical requirements on the juxtaposed RNA ends and the site of ligation, ligation was measured between three TAR variants that carry either a 5′-OH (produced by hammerhead ribozyme cleavage), 5′-monophosphate (by GMP priming), or 5′-triphosphate (natural product of T7 transcription using guanosine triphosphates) and two hairpin oligo variants that carry either a 5′-OH or 5′-monophosphate (5′-triphosphate oligos not commercially available) (Supplementary Fig. 9a). As expected, the TAR 5′-monophosphate end is required to ligate with a hairpin oligo that carry no phosphates, verifying the intended site of ligation. Second, to ensure the specificity of this bimolecular base-pairing interaction, we introduced 1, 2, or 3 mismatches into the 5′-ACC overhang of the probing hairpin. Increasing numbers of mismatches in the 3-bp intermolecular helix progressively reduced ligation efficiency. While one mismatch is tolerated, two or more mismatches blocked ligation (Fig. 5b). A blunt-end hairpin also exhibited modest ligation (7% at 16 h), more than the hairpins bearing mismatched overhangs, potentially due to transient end-to-end stacking between both blunt ends of TAR and the hairpin probe. Lastly, we compared the accessibilities of the TAR 5′ and 3′ ends to hairpin ligation. In contrast to the robust ligation of a 5′-ACC overhang hairpin to TAR 5′ end, a 5′-phosphorylated hairpin probe bearing a 3′-GGU overhang complementary to TAR 3′-ACC end is unable to ligate with TAR (Fig. 5c). This inability to ligate to the TAR 3′-end is likely attributable to the recessed 5′-phosphate on the hairpin probe, which is expected to be more difficult to be adenylated by Rnl2 due to steric hinderance.

### NC-mediated annealing of TAR to cTAR

100 nM of TAR RNA was mixed with 400 nM of complementary DNA cTAR in 25 mM Tris-HCl pH 7.5, 100 mM NaCl in the presence or absence of 0.5 mM $MgCl_2$ in a final volume of 20 μL. 0.9 μM of NC was added to the mixture to reach a ratio of one NC per 6 nucleotides and incubated at 37 °C for different time lengths. The reactions were quenched by adding 1 μL of 10 mg/mL Proteinase K for 10 min at 37 °C to degrade the NC protein while retaining nucleic acid structures. 10 μL of non-denaturing loading buffer (100 mM Tris-HCl pH 7.5, 20% v/v glycerol, 0.025% w/v xylene cyanol, and 0.025% bromophenol blue) was added to the mixture and analyzed on a non-denaturing 10% polyacrylamide (29:1 acrylamide:bisacrylamide) gel containing 20 mM $MgCl_2$. The RNA band intensities were quantified using ImageJ and the data were fit with a single exponential decay to derive apparent rates of annealing $k_{obs}$ $(min^{-1})$.

### Reporting summary

Further information on research design is available in the Nature Portfolio Reporting Summary linked to this article.

## Data availability

The atomic coordinates and structure factor amplitudes for the TAR-I, TAR-II, TAR-I + Tat, TAR-II + $Ca^{2+}$ crystal structures have been deposited at the Protein Data Bank (PDB) under accession codes 9DE6, 9DE7, 9DE5, and 9DE8, respectively. Source data are provided in this paper.

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

## Acknowledgements

We thank H. Abou Assi, C. Chu, and H.M. Al-Hashimi for NMR analyses, I. Botos for computational support, G. Piszczek and D. Wu for support in biophysical measurements, and J.L. Smith, V.M. D'Souza, M.F. Summers, I. Skeparnias, A. Umuhire Juru for discussions. X-ray diffraction data were collected at the Southeast Regional Collaborative Access Team (SER-CAT) 22-ID beamline at the Advances Photon Source of the Argonne National Laboratory, supported by the U.S. Department of Energy under Contract No. W-31–109-Eng-38. This work was supported by the intramural research programs of NIDDK and NHLBI, NIH (ZIADK075136 to J.Z., ZIAHL001452 to J.R.K.), an NIH Deputy Director for Intramural Research (DDIR) Challenge Award to J.Z., and Center for Structural Biology of HIV-1 RNA (CRNA) supported by NIAID U54 AI17660. C.B.N. is a recipient of an Intramural AIDS Research Fellowship (IARF) funded by the NIH Office of AIDS Research, a NIDDK Nancy Nossal Fellowship Award, and an NIAID Pathway to Independence (K99/R00) Award.

## Author contributions

C.B.-N. and J.Z. conceived the study and designed the RNA constructs. C.B.-N. prepared all crystals and collected diffraction data. C.B.-N. and J.Z. determined, refined, and analyzed the structures. C.B.-N. performed FP, ITC, CD, DSC, ligation, and kinase experiments. C.B.-N., K.A.L., J.R.K., and K.C.S. performed and analyzed fluorescence lifetime measurements. All authors contributed to interpreting the data and writing the paper.

## Funding

## Competing interests

The authors declare no competing interests.
