## [Transparent Peer Review file · Nature Communications]

Structures of complete HIV-1 TAR RNA portray a dynamic platform poised for protein binding and structural remodeling

Corresponding Author: Dr Jinwei Zhang

Version 0:

Reviewer comments:

Reviewer #1

(Remarks to the Author)

This manuscript presents four crystal structures of the 57-nucleotide HIV-1 TAR, offering crucial insights into the structures of TAR and its function in Tat binding and PKR inhibition. This is a much-needed study for an important problem in HIV RNA structural biology. Previous structural studies have focused on the TAR RNA's upper stem-loop; this study focused on the lower stem and its function. Important findings from the study include that TAR exhibits a metastable state and significant conformational mobility within three bulges and the lower stem; moreover, TAR inhibits PKR via its lower stem, which shows transient conformational excursions. The manuscript further includes results on RNA overhang analysis, as well as the local and global remodeling of TAR induced by HIV-1 NC, Tat, and PKR.

The work is robust and a tour de force, and the paper is well-organized with detailed descriptions of the findings. The following lists a few minor comments:

1. Line 126. It would be useful to employ a computational model to test whether the three mismatches (A48U, C49G, U50A) would induce the refolding of the lower stem to form new base pairs or bulges.
2. Line 142-166. The number of 5' guanines could modulate TAR's interactions with the adjacent polyA hairpin, which I believe depends on the stability of the lower stem of TAR. The authors' experiments on TAR2G and TAR3GC could provide valuable insights, and it would be useful to show the results in the main manuscript rather than in the supplementary material.
3. Figure 2. C5 exhibits three distinct conformations across the three structures. Please highlight/label C5 in the three 3D structures (using red, as in the 2D structure) to assist readers in identifying it.
4. Line 335-336. Through ROALHO, the authors found that the larger UCU bulge located 18-bp away from the termini exhibits a larger impact to lower stem stability than the much closer single-nucleotide bulges. A further discussion about the possible mechanisms would be helpful.
5. Are the structures and sequences of the lower stem conserved among other HIV-1 variants? For instance, in HIV-1MAL, a TAR with four bulges has been identified (Yasin, Saif, et al, 2024); could this change the TAR function observed in this study?
6. Line 100-102. The figure citations in this paragraph are confusing. The data for the 79 bp dsRNA is presented in Fig. 1b; why is Fig. 1c referenced instead of Fig. 1b in line 102?

Reviewer #2

(Remarks to the Author)

I co-reviewed this manuscript with one of the reviewers who provided the listed reports. This is part of the Nature

Communications initiative to facilitate training in peer review and to provide appropriate recognition for Early Career Researchers who co-review manuscripts.

Reviewer #3

(Remarks to the Author)

The manuscript "Structures of complete HIV-1 TAR RNA portray a dynamic platform poised for protein binding and structural remodeling" by Bou-Nader et al. present structure-function studies of full-length TAR RNA.

- i) The authors begin by performing activation assays of PKR, which demonstrates that TAR inhibits PKR at a similar level to Adenovirus Virus-Associated RNA-1, a known PKR inhibitor.
- ii) The authors then determined that PKR primarily binds to the lower stem, and by systematic elimination of base pairs and mutations that eliminate bulges show that bulges from both the top and bottom stem affect PKR inhibition.
- iii) They use three different TAR constructs and unfolding patterns to demonstrate cooperativity between the upper and lower halves, different abilities to base pair with NC-mediated cTAR, and different inhibitions of PKR.
- iv) The authors then present crystal structures of three different full-length TAR constructs, which demonstrate heterogeneity particularly in the bulges and lower stem. The crystal structure of full-length TAR with the Tat peptide revealed that Tat forms an alpha helix, which is different than a recently solved NMR structure, but show some major features to be similar.
- v) The authors also demonstrate using ROALHO that the lower TAR stem is metastable. Finally, the authors demonstrate through 2AP time-resolved fluorescence spectroscopy that the entire TAR stem is dynamically remodeled upon binding of NC, Tat RBD, and dsRBMs from PKR.

Thus, the authors do several experiments that highlight the crosstalk between the upper and lower stem of TAR and the need for both structural and functional studies of the full-length TAR. There are a few functional aspects and discussion clarifications that need to be addressed to strengthen this work:

- 1) There seems to be inconsistency (or poor description) as to which constructs are being used for studies across the paper, making it difficult to discern whether the conclusions made can truly be broadly applicable across the study:
 - a) For example, it seems that a 2G construct with an overhang is used for Fig. 1 (although this is not clarified).
 - b) The structures that were solved are in a 2G construct with an altered loop along with seemingly the structure-based ITC and ROALHO experiments in Fig. 2-5 (Is the loop also altered in the ROALHO and ITC experiments?).
 - c) In Fig. 6, it seems that TAR with a 2G start but with a G-C/G-U lower stem and WT loop is primarily utilized (this is also not clarified but perhaps may be a mistake with the G-U stem?).
 - d) The 2G construct that seems to be primarily utilized in these studies can indeed be a relevant construct as HIV can form a 1G, 2G, and 3G form that have been shown to, at least at length in the 1G versus 3G context, have different structural and functional consequences. Introduction and discussion points about these structure-function differences, which also affects TAR structures should be included in this manuscript.
 - e) The 3G and 3GC constructs are included in the studies to compare against 2GC constructs to address TSS heterogeneity. While, this study does a good job in addressing that there is structural cooperativity across the upper and the lower stem and demonstrating that 2GC is different than other TSS heterogeneous constructs, additional studies on PKR activation, ROALHO, and 2AP time-resolved fluorescence spectroscopy should be done in 1G, 2G, and 3G contexts with a 5' cap so that it can be understood how the 2GC structure-function studies (which should be consistent across all studies in the manuscript) compare with these physiologically relevant constructs. If the 5' cap constructs are difficult to synthesize, the authors should provide background information to show that the constructs used form similar structures with the 5' cap. Such comparative analysis should be included in the introduction and/or discussion. Furthermore, comparisons with the NMR structures will probably then be more appropriate in these sections as well.
- 2) The paper seems to be organized where the researchers first show that the lower stem has structural and dynamic properties that affect function, which warrant further structural investigations of full-length TAR (which is done in Fig. 2-4 and 6). As such, as it is currently written, it seems more appropriate that the ROALHO experiments with the addition of 1G, 2G, and 3G with a 5' cap should be moved to Fig. 2.

Reviewer #4

(Remarks to the Author)

Version 1:

Reviewer comments:

Reviewer #1

(Remarks to the Author)

The authors have addressed our concerns in a satisfactory way.

Reviewer #2

(Remarks to the Author)

Reviewer #3

(Remarks to the Author)

Our concerns have been appropriately addressed.

Reviewer #4

(Remarks to the Author)

Point-by-point responses to reviewer comments:

Reviewer #1 (Remarks to the Author):

This manuscript presents four crystal structures of the 57-nucleotide HIV-1 TAR, offering crucial insights into the structures of TAR and its function in Tat binding and PKR inhibition. This is a much-needed study for an important problem in HIV RNA structural biology. Previous structural studies have focused on the TAR RNA's upper stem-loop; this study focused on the lower stem and its function. Important findings from the study include that TAR exhibits a metastable state and significant conformational mobility within three bulges and the lower stem; moreover, TAR inhibits PKR via its lower stem, which shows transient conformational excursions. The manuscript further includes results on RNA overhang analysis, as well as the local and global remodeling of TAR induced by HIV-1 NC, Tat, and PKR.

The work is robust and a tour de force, and the paper is well-organized with detailed descriptions of the findings. The following lists a few minor comments:

A: We thank the reviewer for their favorable assessments and helpful suggestions.

1. Line 126. It would be useful to employ a computational model to test whether the three mismatches (A48U, C49G, U50A) would induce the refolding of the lower stem to form new base pairs or bulges.

A: Indeed, we concur with the reviewer that it is important to check if the designed mismatches might induce the formation of alternative, unplanned secondary structures. As suggested, we performed Mfold analyses on the WT (2GC) and A48U/C49G/U50A mutant designed to form a 3-nucleotide internal loop. As shown below, Mfold suggests a single conformer for the A48U/C49G/U50A mutant, yielding a 3-nucleotide internal loop as designed. This computational analysis confirms that the mutant TAR RNA is likely forming the intended secondary structure. We have added this new analysis and figure in Supplementary Fig. 1k.

2. Line 142-166. The number of 5' guanines could modulate TAR's interactions with the adjacent polyA hairpin, which I believe depends on the stability of the lower stem of TAR. The authors' experiments on TAR2G and TAR3GC could provide valuable insights, and it would be useful to show the results in the main manuscript rather than in the supplementary material.

A: We agree with the reviewer's interpretation and appreciate their positive assessment of these data. We made several attempts to move these data to the main figures but found we simply do not have sufficient space to accommodate them, without excessively shrinking their sizes. Therefore, we propose to keep these data in the supplementary material, where we could maximize the dimensions and therefore visibility of these data figures.

3. Figure 2. C5 exhibits three distinct conformations across the three structures. Please highlight/label C5 in the three 3D structures (using red, as in the 2D structure) to assist readers in identifying it.

A: Changed the color of C5 to red in 3D structures as suggested, in Fig. 2a, b, c, g, h, and i.

4. Line 335-336. Through ROALHO, the authors found that the larger UCU bulge located 18-bp away from the termini exhibits a larger impact to lower stem stability than the much closer single-nucleotide bulges. A further discussion about the possible mechanisms would be helpful.

A: As suggested, additional discussion has been added. We now state:

“As the dsRNA grows in length, its overall stacking strength and stability increases accumulatively, thus allowing distant structural features such as bulges to modulate the thermodynamic behavior of the termini. Similar long-range effects were previously observed in the T-box riboswitch-tRNA complex, where the overall stability of a 33-bp-long “central spine” assembled from four coaxially arranged dsRNA segments determines the regulatory outcome of the riboswitch^{59,60}.

5. Are the structures and sequences of the lower stem conserved among other HIV-1 variants? For instance, in HIV-1MAL, a TAR with four bulges has been identified (Yasin, Saif, et al, 2024); could this change the TAR function observed in this study?

A: The reviewer brings up an excellent point. Yes, the lower stem of TAR is about as conserved as the upper stem, as indicated below (left) by the Rfam conservation analysis (<http://rfam.org/family/RF00250>). The fact that HIV-1 MAL strain has 4 bulges is indeed curious (right, below). The addition of a 4th bulge at U8 coincides with the reduction of the UCU bulge size from 3 to 2 nts. The alteration of the bulges in the MAL strain is consistent with the notion in our manuscript that the lower stem of TAR needs to remain dynamic and imperfectly paired. It is possible that the additional bulge in HIV MAL confers even higher structural dynamics of the TAR termini than the standard NL4-3 strain used here, which may have better suited the viral habits of the particular strain. We have added this interesting observation to the text and the reference. We now state:

“Curiously, the HIV-1 MAL strain employs a TAR element that features one additional single-nt bulge (U8) besides the conserved C5 bulge in the lower stem.”

6. Line 100-102. The figure citations in this paragraph are confusing. The data for the 79 bp dsRNA is presented in Fig. 1b; why is Fig. 1c referenced instead of Fig. 1b in line 102?

A: To clarify, Fig. 1b is the PKR activation assay, while Fig. 1c depicts the competitive PKR inhibition assay, where both the 79-bp dsRNA and viral RNA were present. It is Fig. 1c that shows TAR, when prebound to PKR, prevents the competition and activation by the 79 bp dsRNA added later, in a dosage-dependent manner.

Minor comments:

1. Some wording in the text could be rephrased.

A: We have now carefully proofread and smoothed the language in the text where needed.

Reviewer #2 (Remarks to the Author):

A: We support and thank the reviewer for participating in the Early Career review program.

Reviewer #3 (Remarks to the Author):

The manuscript “Structures of complete HIV-1 TAR RNA portray a dynamic platform poised for protein binding and structural remodeling” by Bou-Nader et al. present structure-function studies of full-length TAR RNA.

i) The authors begin by performing activation assays of PKR, which demonstrates that TAR inhibits PKR at a similar level to Adenovirus Virus-Associated RNA-1, a known PKR inhibitor.

ii) The authors then determined that PKR primarily binds to the lower stem, and by systematic elimination of base pairs and mutations that eliminate bulges show that bulges from both the top and bottom stem affect PKR inhibition.

iii) They use three different TAR constructs and unfolding patterns to demonstrate cooperativity between the upper and lower halves, different abilities to base pair with NC-mediated cTAR, and

different inhibitions of PKR.

iv) The authors then present crystal structures of three different full-length TAR constructs, which demonstrate heterogeneity particularly in the bulges and lower stem. The crystal structure of full-length TAR with the Tat peptide revealed that Tat forms an alpha helix, which is different than a recently solved NMR structure, but show some major features to be similar.

v) The authors also demonstrate using ROALHO that the lower TAR stem is metastable. Finally, the authors demonstrate through 2AP time-resolved fluorescence spectroscopy that the entire TAR stem is dynamically remodeled upon binding of NC, Tat RBD, and dsRBMs from PKR.

Thus, the authors do several experiments that highlight the crosstalk between the upper and lower stem of TAR and the need for both structural and functional studies of the full-length TAR. There are a few functional aspects and discussion clarifications that need to be addressed to strengthen this work:

A: We thank the reviewer for their detailed analyses, interpretations and helpful suggestions for strengthening our manuscript.

1) There seems to be inconsistency (or poor description) as to which constructs are being used for studies across the paper, making it difficult to discern whether the conclusions made can truly be broadly applicable across the study:

A: Indeed, the nomenclature can quickly get complex and confusing, due to the natural 5' end heterogeneity of HIV-1 TAR RNA, and inconsistent naming schemes used in historic and current literature. To clarify exactly which RNA structure is being used, we have now explicitly defined the 5' end status for each experiment both in the figures and in their legends. Where applicable, we have added miniaturized secondary structure insets and/or additional text labels in the figures to more clearly show the 5' ends of the TAR constructs used in each assay, in Fig. 1b, Fig. 4g, Fig. 4h, and Fig. 5a.

a) For example, it seems that a 2G construct with an overhang is used for Fig. 1 (although this is not clarified).

A: Like in most other figures, Fig. 1 used the blunt end 2GC construct, which we defined as "WT" in the context of this manuscript, in line with the prevailing construct used in most TAR literature. We grayed out the G1 residue to indicate its absence, which we now explicitly define. Also, as shown in Fig. S3F, TAR 5' heterogeneities had little impact on the inhibitory activity against PKR. We have now clarified the exact construct used in all Fig. 1 panels and the legend.

b) The structures that were solved are in a 2G construct with an altered loop along with seemingly the structure-based ITC and ROALHO experiments in Fig. 2-5 (Is the loop also altered in the ROALHO and ITC experiments?).

A: Yes, both the structural analyses and biochemical analyses primarily used the 2GC construct. In the ROALHO and ITC experiments, we used 2GC TAR RNAs that bear WT apical loops, to avoid any potential interference of the engineered apical loop in these experiments. We have now clarified this in the method section. Now we state:

"All TAR RNAs used in ROALHO bear the natural hexanucleotide (CUGGGA) apical loop."

c) In Fig. 6, it seems that TAR with a 2G start but with a G-C/G-U lower stem and WT loop is primarily utilized (this is also not clarified but perhaps may be a mistake with the G-U stem?).

A: We thank the reviewer for pointing out this issue, which we should have clarified in the original manuscript. Yes the 2G start construct is used throughout Fig. 6. The use of the G•U pair in the penultimate base pair position (i.e., C57U substitution, only in Fig. 6c) instead of G-C is necessitated by the technical requirement to properly base pair it with 2AP, as 2AP-C pairs are unstable and could exhibit unusual and multiple pairing geometries (L C Sowers et al., 2000, *Biochemistry, Multiple Structures for the 2-Aminopurine-Cytosine Mismatch*). This change could have slightly affected the measured lifetimes and dynamics of the region compared to the WT RNA bearing the G-C pair, but is nonetheless unavoidable. We have now explicitly noted this caveat in the main text and figure legend. We now state:

“Notably, due to instability and multiple pairing configurations of the 2AP-C mismatch, we introduced a C57U substitution opposite 2AP at G3 position to ensure Watson-Crick pairing at this location (Fig. 6c, italic). Although this change was necessary, it may have slightly altered the local structural dynamics.”

d) The 2G construct that seems to be primarily utilized in these studies can indeed be a relevant construct as HIV can form a 1G, 2G, and 3G form that have been shown to, at least at length in the 1G versus 3G context, have different structural and functional consequences. Introduction and discussion points about these structure-function differences, which also affects TAR structures should be included in this manuscript.

A: As suggested, we have expanded the introduction and discussion sections to include more information about the 5' heterogeneities and the cap. We now state in Introduction:

“Proviral transcription of the HIV-1 and other retroviral genomes can initiate from any of the three guanosines (Gs) at the 5' end of TAR and is cotranscriptionally capped, resulting in substantial transcription start site (TSS) heterogeneity. This variable number of 5' Gs on TAR controls the conformation and fate of the 5' leader and entire genome, in part by modulating TAR interactions with the immediately adjacent polyA hairpin. While leaders bearing a single 5' G are preferably packaged into virions, 3G leaders are selectively retained in cells and serve as mRNAs for translation. In addition, both types of leaders bear the 7-methylguanosine (m7G) cap, which can further stack with the 5' G(s) in an inverted geometry. While the capped 3G leader extrudes and exposes its 5' cap for translation initiation factor (eIF4E) binding and translation, the capped 1G leader does not bind eIF4E under similar conditions, directing it away from ribosomes. The lower, proximal stem of TAR coordinates with the 5' TSS and the cap to control the overall leader conformation and its fate. In agreement, disruption of the TAR hairpin causes aberrant genome dimerization and packaging. While the hairpin structure of TAR suppresses ribosomal scanning, a TAR-polyA coaxial stacking interaction within the capped 1G leader additionally conceals the cap from eIF4E, thus reinforcing the translation block.”

We now state in Discussion:

“Further, the TAR constructs used in this study lack the 5' cap, which is covalently linked to and can stack with the immediately adjacent 5' G or a blunt end base pair involving the 5' G. It will be of interest to further explore how the presence of the 5' cap, followed by one, two, or three Gs, may modulate the TAR structural dynamics.”

e) The 3G and 3GC constructs are included in the studies to compare against 2GC constructs to address TSS heterogeneity. While, this study does a good job in addressing that there is structural cooperativity across the upper and the lower stem and demonstrating that 2GC is different than other TSS heterogeneous constructs, additional studies on PKR activation, ROALHO, and 2AP time-resolved fluorescence spectroscopy should be done in 1G, 2G, and 3G contexts with a 5' cap so that it can be understood how the 2GC structure-function studies (which should be consistent across all studies in the manuscript) compare with these physiologically relevant constructs. If the 5' cap constructs are difficult to synthesize, the authors should provide background information to show that the constructs used form similar structures with the 5' cap. Such comparative analysis should be included in the introduction and/or discussion. Furthermore, comparisons with the NMR structures will probably then be more appropriate in these sections as well.

A: We agree with the reviewer that testing the 5' cap would be quite interesting. However, unfortunately ROALO is most likely incompatible with 5' capped RNAs, as the inverted chemical structure of the cap is known to prevent ligation (Ho, C. K. 2017., *Methods Mol. Biol.* 1648, 1–9). For PKR assays, we've found that 5' length heterogeneities had minimal impact on PKR inhibition (Fig. S3F). As PKR's dsRBMs are known to recognize the internal dsRNA regions rather than the termini, it seems fairly unlikely that adding the 5' cap (similar to Gs) would drastically impact TAR inhibition of PKR. Indeed, we currently do not have the technical ability and experience to prepare these fairly long RNAs with the natural 5' cap structure. Although attractive, a complete analysis of the precise effects of the 5' heterogeneity and the 5' cap on TAR structural dynamics, best studied when the adjacent polyA is also present, is beyond the scope and intention of our current manuscript. Instead, we have provided additional information, analyses, and discussions, including those of the NMR structures, in both the Introduction and Discussion sections, as suggested by the reviewer. Please also refer to the response to the immediately preceding item.

2) The paper seems to be organized where the researchers first show that the lower stem has structural and dynamic properties that affect function, which warrant further structural investigations of full-length TAR (which is done in Fig. 2-4 and 6). As such, as it is currently written, it seems more appropriate that the ROALHO experiments with the addition of 1G, 2G, and 3G with a 5' cap should be moved to Fig. 2.

A: As mentioned above the 5' cap structure is most likely incompatible with ROALHO as it blocks ligation (Ho, C. K. 2017., *Methods Mol. Biol.* 1648, 1–9). Therefore, these experiments proposed by the reviewer is likely best performed using NMR or other analyses that don't require enzymatic ligation.

Reviewer #4 (Remarks to the Author):

A: We support and thank the reviewer for participating in the Early Career review program.